# Private Geometric Median in Nearly-Linear Time

**Syamantak Kumar**
University of Texas at Austin
syamantak@utexas.edu

**Daogao Liu**
Google Research
liudaogao@gmail.com

**Kevin Tian**
University of Texas at Austin
kjtian@cs.utexas.edu

**Chutong Yang**
University of Texas at Austin
cyang98@cs.utexas.edu

## Abstract

Estimating the geometric median of a dataset is a robust counterpart to mean estimation, and is a fundamental problem in computational geometry. Recently, [HSU24] gave an $(\epsilon, \delta)$-differentially private algorithm obtaining an $\alpha$-multiplicative approximation to the geometric median objective, $\frac{1}{n} \sum_{i \in [n]} \|\cdot - \mathbf{x}_i\|$, given a dataset $\mathcal{D} := \{\mathbf{x}_i\}_{i \in [n]} \subset \mathbb{R}^d$. Their algorithm requires $n \gtrsim \sqrt{d} \cdot \frac{1}{\alpha \epsilon}$ samples, which they prove is information-theoretically optimal. This result is surprising because its error scales with the *effective radius* of $\mathcal{D}$ (i.e., of a ball capturing most points), rather than the worst-case radius. We give an improved algorithm that obtains the same approximation quality, also using $n \gtrsim \sqrt{d} \cdot \frac{1}{\alpha \epsilon}$ samples, but in time $\widetilde{O}(nd + \frac{d}{\alpha^2})$. Our runtime is nearly-linear, plus the cost of the cheapest non-private first-order method due to [CLM+16]. To achieve our results, we use subsampling and geometric aggregation tools inspired by FriendlyCore [TCK+22] to speed up the "warm start" component of the [HSU24] algorithm, combined with a careful custom analysis of DP-SGD's sensitivity for the geometric median objective.

## 1 Introduction

The *geometric median* problem, a.k.a. the Fermat-Weber problem, is one of the oldest problems in computational geometry. In this problem, we are given a dataset $\mathcal{D} = \{\mathbf{x}_i\}_{i \in [n]} \subset \mathbb{R}^d$, and our goal is to find a point $\mathbf{x}_\star \in \mathbb{R}^d$ that minimizes the average Euclidean distance to points in the dataset:

$$\mathbf{x}_\star \in \arg\min_{\mathbf{x} \in \mathbb{R}^d} f_{\mathcal{D}}(\mathbf{x}), \text{ where } f_{\mathcal{D}}(\mathbf{x}) := \frac{1}{n} \sum_{i \in [n]} \|\mathbf{x} - \mathbf{x}_i\|. \tag{1}$$

This problem has received widespread interest due to its applications in high-dimensional statistics. In particular, the geometric median of a dataset $\mathcal{D}$ enjoys robustness properties that the mean (i.e., $\frac{1}{n} \sum_{i \in [n]} \mathbf{x}_i$, the minimizer of $\frac{1}{n} \sum_{i \in [n]} \|\mathbf{x} - \mathbf{x}_i\|^2$) does not. For example, it is known (cf. Lemma 4) that if greater than half of $\mathcal{D}$ lies within a distance $r$ of some $\bar{\mathbf{x}} \in \mathbb{R}^d$, then the geometric median lies within $O(r)$ of $\bar{\mathbf{x}}$. Thus, the geometric median provides strong estimation guarantees even when $\mathcal{D}$ contains outliers. This is in contrast to simpler estimators such as the mean, which can be arbitrarily corrupted by a single outlier. As a result, studying the properties and computational aspects of the geometric median has a long history, see e.g., [Web29, LR91] for some famous examples.

We provide improved algorithms for estimating (1) subject to $(\epsilon, \delta)$-differential privacy (DP, Definition 1), the de facto notion of provable privacy in modern machine learning. Privately computing the geometric median naturally fits into a recent line of work on designing DP algorithms in the presence

of outliers. To explain the challenge of such problems, the definition of DP implies that privacy must hold for *worst-case* datasets. This stringent definition affords DP a variety of desirable properties, most notably *composition* of private mechanisms (cf. [DR14], Section 3.5). However, it also begets challenges: for example, estimating the empirical mean of $\mathcal{D}$ subject to $(\epsilon, \delta)$-DP necessarily results in error scaling $\propto R$, the diameter of the dataset (cf. Section 5, [BST14]). Moreover, the worst-case nature of DP is at odds with typical *average-case* machine learning settings, where most (or all) of $\mathcal{D}$ is drawn from a distribution that we wish to learn about. From an algorithm design standpoint, the question follows: how do we design methods with privacy guarantees for worst-case data, but also with improved utility guarantees for (mostly) average-case data?

Such questions have been successfully addressed for various statistical problems, e.g., parameter estimation [BD14, KV17, BKSW19, DFM+20, BDKU20, BGS+21, AL22, LKJO22, KDH23, BHS23], clustering [NRS07, NSV16, CKM+21, TCK+22], and more. However, existing approaches for (1) (even non-privately) are based on iterative optimization methods, as the geometric median does not admit a simple, closed-form solution. Much of the DP optimization toolkit is exactly plagued by the aforementioned "worst-case sensitivity" issues, e.g., lower bounds for general stochastic optimization problems again scale with the domain size. This is troubling in the context of (1), because a major appeal of the geometric median is its robustness: its error should not be significantly affected by any small subset of the data. Privately estimating the geometric median thus poses an interesting technical challenge, beyond its potential appeal as a subroutine in downstream robust algorithms.

To explain the distinction between worst-case and average-case error rates in the context of (1), we introduce the following helpful notation: for all quantiles $\tau \in [0, 1]$, we let

$$r^{(\tau)} := \arg\min_{r \geq 0} \left\{ \sum_{i \in [n]} \mathbb{1}_{\|\mathbf{x}_i - \mathbf{x}_\star\| \leq r} \geq \tau n \right\}, \text{ where } \mathbf{x}_\star := \arg\min_{\mathbf{x} \in \mathbb{R}^d} \frac{1}{n} \sum_{i \in [n]} \|\mathbf{x} - \mathbf{x}_i\|,$$

$$\text{and } \mathbb{1}_{\|\mathbf{x}_i - \mathbf{x}_\star\| \leq r} = \begin{cases} 1, & \|\mathbf{x}_i - \mathbf{x}_\star\| \leq r \\ 0 & \text{otherwise} \end{cases} \tag{2}$$

when $\mathcal{D} = \{\mathbf{x}_i\}_{i \in [n]} \subset \mathbb{R}^d$ is clear from context. In other words, $r^{(\tau)}$ is the smallest radius describing a ball around the geometric median $\mathbf{x}_\star$ containing at least $\tau n$ points in $\mathcal{D}$. We also use $R$ to denote an a priori overall domain size bound, where we are guaranteed that $\mathcal{D} \subset \mathbb{B}^d(R)$. Note that in general, it is possible for, e.g., $r^{(0.9)} \ll R$ if $\approx 10\%$ of $\mathcal{D}$ consists of outliers with atypical norms. Due to the robust nature of the geometric median (i.e., the aforementioned Lemma 4), a natural target is estimation error scaling with the "effective radius" $r^{(\tau)}$ for some quantile $\tau \in (0.5, 1)$. This is a much stronger guarantee than the error rates $\propto R$ that typical DP optimization methods give.

Because a simple argument (Lemma 3) shows $r^{(\tau)} = O(f_{\mathcal{D}}(\mathbf{x}_\star))$ for all $\tau < 1$, in this introduction our goal is to approximate the minimizer of (1) to additive error $\alpha f_{\mathcal{D}}(\mathbf{x}_\star)$ for some $\alpha \in (0, 1)$, i.e., to give $\alpha$-*multiplicative error* guarantees on optimizing $f_{\mathcal{D}}$.[1] Again, datasets with outliers may have $f_{\mathcal{D}}(\mathbf{x}_\star) \ll R$, so this is beyond the reach of naïvely applying DP optimization methods.

In a recent exciting work, [HSU24] bypassed this obstacle and obtained such private multiplicative approximations to the geometric median, and with near-optimal sample complexity. Assuming that $\mathcal{D}$ has size $n \gtrsim \sqrt{d} \cdot \frac{1}{\alpha\epsilon}$,[2] [HSU24] gave two algorithms for estimating (1) to $\alpha$-multiplicative error (cf. Appendix A). They also proved a matching lower bound, showing this many samples is information-theoretically necessary.[3] From both a theoretical and practical perspective, the main outstanding question left by [HSU24] is computational efficiency: in particular, the [HSU24] algorithms ran in time $\widetilde{O}(n^2 d + n^3 \epsilon^2)$ or $\widetilde{O}(n^2 d + n d^2 + d^{4.372})$. This leaves a significant gap between algorithms for privately solving (1), and their counterparts in the non-private setting, where [CLM+16] showed that (1) could be approximated to $\alpha$-multiplicative error in nearly-linear time $\widetilde{O}(\min(nd, \frac{d}{\alpha^2}))$.

---

[1]Our results (and those of [HSU24]) in fact give stronger additive error bounds of $\alpha r^{(\tau)}$ for any $\tau \in (0.5, 1)$.

[2]In this introduction only, we use $\widetilde{O}, \lesssim, \gtrsim$ to hide polylogarithmic factors in problem parameters, i.e., $d, \frac{1}{\alpha}$, $\frac{1}{\epsilon}, \frac{1}{\delta}$, and $\frac{R}{r}$, where $\mathcal{D} \subseteq \mathbb{B}^d(R)$ and $r \leq r^{(0.9)}$. Our formal theorem statements explicitly state all dependences.

[3]Intuitively, we require $\alpha \approx d^{-1/2}$ to obtain nontrivial estimation when $\mathcal{D}$ consists of i.i.d. Gaussian data (as a typical radius is $\approx \sqrt{d}$), matching known lower bounds of $n \approx \frac{d}{\epsilon}$ for Gaussian mean estimation [KLSU19].

## 1.1 Our results

Our main contribution is a faster algorithm for privately approximating (1) to $\alpha$-multiplicative error.

**Theorem 1** (informal, see Theorem 4). *Let $\mathcal{D} = \{\mathbf{x}_i\}_{i \in [n]} \subset \mathbb{B}^d(R)$ for $R > 0$, $0 < r \leq r^{(0.9)}$, and $(\alpha, \epsilon, \delta) \in [0, 1]^3$. There is an $(\epsilon, \delta)$-DP algorithm that returns $\hat{\mathbf{x}}$ such that with probability $\geq 1 - \delta$, $f_{\mathcal{D}}(\hat{\mathbf{x}}) \leq (1 + \alpha) f_{\mathcal{D}}(\mathbf{x}_\star)$, assuming $n \gtrsim \frac{\sqrt{d}}{\alpha \epsilon}$. The algorithm runs in time $\widetilde{O}(nd + \frac{d}{\alpha^2})$.*

To briefly explain Theorem 1's statement, it uses a priori knowledge of $0 < r < R$ such that $R$ upper bounds the domain size of $\mathcal{D}$, and $r$ lower bounds the "effective radius" $r^{(0.9)}$. However, its runtime only depends polylogarithmically on the aspect ratio $\frac{R}{r}$, rather than polynomially (as naïve DP optimization methods would); we also remark that our sample complexity is independent of $\frac{R}{r}$.

The runtime of Theorem 1 is nearly-linear when $n \gtrsim \frac{1}{\alpha^2}$ (e.g., if $\sqrt{d} \cdot \frac{1}{\epsilon} \gtrsim \frac{1}{\alpha}$), but more generally it does incur an additive overhead of $\frac{d}{\alpha^2}$. This overhead matches the fastest non-private first-order method for approximating (1) to $\alpha$-multiplicative error, due to [CLM+16]. We note that [CLM+16] also gave a second-order interior-point method, that non-privately solves (1) in time $\widetilde{O}(nd)$, i.e., with polylogarithmic dependence on $\frac{1}{\alpha}$. We leave removing this additive runtime term in the DP setting, or proving this is impossible in concrete query models, as a challenging question for future work.

Our algorithm follows a roadmap by [HSU24], who split their algorithm into two phases: an initial "warm start" phase that computes an $O(1)$-multiplicative approximation of the geometric median, and a secondary "boosting" phase that uses iterative methods to improve the warm start to an $\alpha$-multiplicative approximation. The warm start improves the domain size of the boosting phase to scale with the effective radius. However, both the warm start and the boosting phases of [HSU24] required superlinear $\approx n^2 d$ time. Our improvement to the warm start phase of the [HSU24] is quite simple, and may be of independent interest, so we provide a self-contained statement here.

**Theorem 2** (informal, see Theorem 3). *Let $\mathcal{D} = \{\mathbf{x}_i\}_{i \in [n]} \subset \mathbb{B}^d(R)$ for $R > 0$, $0 < r \leq r^{(0.9)}$, and $(\epsilon, \delta) \in [0, 1]^2$. There is an $(\epsilon, \delta)$-DP algorithm that returns $\hat{\mathbf{x}}$ such that with probability $\geq 1 - \delta$, $f_{\mathcal{D}}(\hat{\mathbf{x}}) = O(f_{\mathcal{D}}(\mathbf{x}_\star))$, assuming $n \gtrsim \frac{\sqrt{d}}{\epsilon}$. The algorithm runs in time $\widetilde{O}(nd)$.*

## 1.2 Our techniques

As discussed previously, our algorithm employs a similar framework as [HSU24]. It is convenient to further split the warm start phase of the algorithm into two parts: finding an estimate $\hat{r}$ of the effective radius of $\mathcal{D}$, and finding an approximate centerpoint at distance $O(\hat{r})$ from the geometric median $\mathbf{x}_\star$.

**Radius estimation.** Our radius estimation algorithm is almost identical to that in [HSU24], Section 2.1, which uses the sparse vector technique (cf. Lemma 2) to detect the first time an estimate $\hat{r}$ is such that most points have $\geq \frac{3}{4}$ of $\mathcal{D}$ at a distance of $\approx \hat{r}$. The estimate $\hat{r}$ is geometrically updated over a grid of size $O(\log(\frac{R}{r}))$. Naïvely implemented, this strategy takes $\gtrsim n^2 d$ time due to the need for pairwise distance comparisons (cf. Appendix A); even if dimensionality reduction techniques are used, this step appears to require $\Omega(n^2)$ time. We make a simple observation that a random sample of $\approx \log(\frac{1}{\delta})$ points from $\mathcal{D}$ is enough to determine whether a given point has $\gg \beta$ neighbors, or $\ll \gamma$, for appropriate (constant) quantile thresholds $\beta, \gamma$, which is enough to obtain an $\widetilde{O}(nd)$ runtime.

**Centerpoint estimation.** Our centerpoint estimation step departs from [HSU24], Section 2.2, who analyzed a custom variant of DP gradient descent with geometrically-decaying step sizes. We make the simple observation that directly applying the FriendlyCore algorithm of [TCK+22] yields the same result. However, the standard implementation of FriendlyCore again uses $\Omega(n^2)$ time to estimate weights for each data point. We show that FriendlyCore can be sped up to run in $\widetilde{O}(nd)$ time (independently of $\frac{R}{r}$) using subsampled weights. Our privacy proof is subtle, and based on an argument (Lemma 1) that couples our subsampled algorithm to an idealized algorithm that never fails to be private. We use this to account for the privacy loss due to the failure of our subsampling, i.e., if the estimates are inaccurate. We note that the [HSU24] algorithm for this step already ran in nearly-linear $\approx nd \log(\frac{R}{r})$ time, so we obtain an asymptotic improvement only if $\frac{R}{r}$ is large.

**Boosting.** The most technically novel part of our algorithm is in the boosting phase, which takes as input a radius and centerpoint estimate from the previous steps, and outputs an $\alpha$-multiplicative

approximation to (1). Like [HSU24], we use iterative methods to implement this phase. However, a major bottleneck to a faster algorithm is the lack of a nearly-linear time DP solver for non-smooth empirical risk minimization (ERM) problems. Indeed, such $\widetilde{O}(1)$-pass optimizers are known only when the objective is convex and sufficiently smooth [FKT20], or $n \gtrsim d^2$ samples are taken [CJJ$^+$23]. This is an issue, because while computing the geometric median (1) is a convex ERM problem, it is non-smooth, and multiplicative guarantees are possible even with $n \approx \sqrt{d}$ samples.

We give a custom analysis of DP-SGD, specifically catered to the (non-smooth) ERM objective (1). Our main contribution is a tighter sensitivity analysis of DP-SGD's iterates, leveraging the structure of the geometric median. To motivate this observation, consider coupled algorithms with iterates $\mathbf{z}, \mathbf{z}'$, both taking gradient steps with respect to the subsampled function $\|\cdot - \mathbf{x}_i\|$ for some dataset element $\mathbf{x}_i \in \mathcal{D}$. A simple calculation (7) shows these gradients are unit vectors $\mathbf{u}, \mathbf{u}'$, in the directions of $\mathbf{z} - \mathbf{x}_i$ and $\mathbf{z}' - \mathbf{x}_i$ respectively. It is not hard to formalize (Lemma 11) that updating $\mathbf{z} \leftarrow \mathbf{z} - \eta\mathbf{u}$ and $\mathbf{z}' \leftarrow \mathbf{z}' - \eta\mathbf{u}'$ is always contractive, unless $\mathbf{z}, \mathbf{z}'$ were both already very close to $\mathbf{x}_i$ (and hence, each other) to begin with. We use this structural result to inductively control DP-SGD's sensitivity, which lets us leverage a prior reduction from private optimization to stable optimization [FKT20].

Our result is the first we are aware of that obtains a nearly-linear runtime for DP-SGD on a structured non-smooth problem. We were inspired by [ALT24], who also gave faster runtimes for (smooth) DP optimization problems with outliers under further assumptions on the objective. We hope that our work motivates future DP optimization methods that harness problem structure for improved rates.

### 1.3 Related work

**Differentially private convex optimization.** DP convex optimization has been studied extensively for over a decade [CM08, KST12, BST14, KJ16, BFGT20, FKT20, BGN21, GLL22, GLL$^+$23] and inspired the influential DP-SGD algorithm widely adopted in deep learning [ACG$^+$16]. In the classic setting, where functions are assumed to be Lipschitz and defined over a convex domain of diameter $R$, optimal rates have been achieved with linear dependence on $R$ [BFTGT19]. Recent years have seen significant advancements in the gradient complexity of DP stochastic convex optimization [FKT20, AFKT21, KLL21, ZTC22, CJJ$^+$23, CCGT24]. Despite these efforts, a nearly-linear gradient complexity has only been established for sufficiently smooth functions [FKT20, ZTC22, CCGT24] and for non-smooth functions [CJJ$^+$23] when $\sqrt{n} \gtrsim d$ is satisfied. Finally, in a related direction, [KMM$^+$25] study private convex hull estimation and it remains an interesting future direction to see if our techniques apply to their setting. As noted in their work, the convex hull can be a very unstable function of the dataset, whereas multiplicative approximations to the geometric median are not affected drastically by a single datapoint (e.g., Lemma 4).

**Differential privacy with average-case data.** Adapting noise to the inherent properties of data, rather than catering to worst-case scenarios, is critical for making differential privacy practical in real-world applications. Several important approaches have emerged in this direction: smooth sensitivity frameworks [NRS07] that refine local sensitivity to make it private; instance optimality techniques [AD20] with tailored guarantees for specific datasets; dmethods with improved performance under distributional assumptions such as sub-Gaussian or heavy-tailed i.i.d. data [CWZ21, AL23, ALT24]; and data-dependent sensitivity computations that adapt during algorithm execution [ATMR21]. These approaches collectively represent the frontier in balancing privacy and utility beyond worst-case analyses. We view our work as another contribution towards this broader program.

## 2 Preliminaries

Throughout, vectors are denoted in lowercase boldface, and the all-zeroes and all-ones vectors in dimension $d$ are respectively $\mathbf{0}_d$ and $\mathbf{1}_d$. We use $\|\cdot\|$ to denote the Euclidean ($\ell_2$) norm of a vector. We use $[d]$ to denote $\{i \in \mathbb{N} \mid 1 \le i \le d\}$. We use $\mathbb{B}^d(\boldsymbol{\mu}, r) := \{\mathbf{x} \in \mathbb{R}^d \mid \|\mathbf{x} - \boldsymbol{\mu}\| \le r\}$ to denote the Euclidean ball of radius $r > 0$ around $\boldsymbol{\mu} \in \mathbb{R}^d$; when $\boldsymbol{\mu}$ is unspecified, then $\boldsymbol{\mu} = \mathbf{0}_d$ by default. For compact $\mathcal{K} \subseteq \mathbb{R}^d$, we use $\mathbf{\Pi}_{\mathcal{K}}(\mathbf{x})$ to denote the Euclidean projection $\arg\min_{\mathbf{y} \in \mathcal{K}} \|\mathbf{x} - \mathbf{y}\|$.

We let $\mathbb{I}_{\mathcal{E}}$ denote the 0-1 indicator random variable of an event $\mathcal{E}$. For two densities $\mu, \nu$ on the same sample space $\Omega$ and $\alpha > 1$, we define the $\alpha$-Rényi divergence $D_\alpha(\mu\|\nu) := \frac{1}{\alpha-1} \log(\int (\frac{\mu(\omega)}{\nu(\omega)})^\alpha \nu(\omega)\mathrm{d}\omega)$. We use $\mathcal{N}(\boldsymbol{\mu}, \sigma^2\mathbf{I}_d)$ to denote the multivariate normal distribution

with mean $\boldsymbol{\mu} \in \mathbb{R}^d$ and covariance $\sigma^2 \mathbf{I}_d$, where $\mathbf{I}_d$ is the $d \times d$ identity. We let $\mathsf{Laplace}(\lambda)$ be the Laplace distribution with scale parameter $\lambda \geq 0$, whose density is $\propto \exp(-\frac{|\cdot|}{\lambda})$. We let $\mathsf{Unif}(S)$ denote the uniform distribution over a set $S$, and $\mathsf{Bern}(p)$ denote the Bernoulli distribution taking on values $\{0, 1\}$ with mean $p \in [0, 1]$. We refer to a product distribution consisting of $k$ i.i.d. copies of a base distribution $\mathcal{D}$ by $\mathcal{D}^{\otimes k}$. We also use the bounded Laplace distribution with parameters $\lambda, \tau \geq 0$, denoted $\mathsf{BoundedLaplace}(\lambda, \tau)$, i.e., the distribution of $X \sim \mathsf{Laplace}(\lambda)$ conditioned on $|X| \leq \tau$.

**Differential privacy.** Let $\mathcal{X}$ be some domain, and let $\mathcal{D} \in \mathcal{X}^n$ be a dataset consisting of $n$ elements from $\mathcal{X}$. We say datasets $\mathcal{D}, \mathcal{D}' \in \mathcal{X}^n$ are *neighboring* if their symmetric difference has size 1, i.e., they differ in a single element. We use the following definition of differential privacy in this paper.

**Definition 1** (Differential privacy). *Let* $(\epsilon, \delta) \in [0, 1]^2$.[4] *We say that a randomized algorithm* $\mathcal{A} : \mathcal{X}^n \to \Omega$ *satisfies* $(\epsilon, \delta)$*-differential privacy (or, is* $(\epsilon, \delta)$*-DP) if for all events* $\mathcal{E} \subseteq \Omega$*, and for all neighboring datasets* $\mathcal{D}, \mathcal{D}' \in \mathcal{X}^n$*,* $\Pr[\mathcal{A}(\mathcal{D}) \in \mathcal{E}] \leq \exp(\epsilon) \Pr[\mathcal{A}(\mathcal{D}') \in \mathcal{E}] + \delta$.

DP algorithms satisfy *basic composition* (Theorem B.1, [DR14]), i.e., if $\mathcal{A}_1 : \mathcal{X}^n \to \Omega_1$ is $(\epsilon_1, \delta_1)$-DP and $\mathcal{A}_2 : \mathcal{X}^n \times \Omega_1 \to \Omega_2$ is $(\epsilon_2, \delta_2)$-DP, then their composition is $(\epsilon_1 + \epsilon_2, \delta_1 + \delta_2)$-DP. We next state the Gaussian mechanism. Recall that if $\mathbf{v} : \mathcal{X}^n \to \mathbb{R}^k$ is a vector-valued function of a dataset, we say $\mathbf{v}$ has sensitivity $\Delta$ if for all neighboring $\mathcal{D}, \mathcal{D}' \in \mathcal{X}^n$, $\|\mathbf{v}(\mathcal{D}) - \mathbf{v}(\mathcal{D}')\| \leq \Delta$.

**Fact 1** (Theorem A.1, [DR14]). *Let* $\mathbf{v} : \mathcal{X}^n \to \mathbb{R}^k$ *have sensitivity* $\Delta$*, and let* $(\epsilon, \delta) \in [0, 1]^2$*. Then, drawing a sample from* $\mathcal{N}(\mathbf{v}(\mathcal{D}), \sigma^2 \mathbf{I}_k)$ *is* $(\epsilon, \delta)$*-DP, for any* $\sigma \geq \frac{2\Delta}{\epsilon} \cdot \sqrt{\log(\frac{2}{\delta})}$.

We also require the bounded Laplace mechanism, which is known to give the following guarantee.

**Fact 2** (Lemma 9, [ALT24]). *Let* $s : \mathcal{X}^n \to \mathbb{R}$ *have sensitivity* $\Delta$*, and let* $(\epsilon, \delta) \in [0, 1]^2$*. Then, drawing* $\xi \sim \mathsf{BoundedLaplace}(\frac{\Delta}{\epsilon}, \tau)$ *and outputting* $s(\mathcal{D}) + \xi$ *is* $(\epsilon, \delta)$*-DP for any* $\tau \geq \frac{\Delta}{\epsilon} \log(\frac{4}{\delta})$.

Fact 2 is proven in [ALT24] using a coupling argument, using the fact that $\mathsf{BoundedLaplace}(\lambda)$ and $\mathsf{Laplace}(\lambda)$ result in the same sample except with some probability. We appeal to this technique several times in Section 3, so we explicitly state it here for convenience (with a proof in Appendix B).

**Lemma 1.** *For* $(\epsilon, \delta) \in [0, 1]^2$*,* $\mathcal{A} : \mathcal{X}^n \to \Omega$ *be an* $(\epsilon, \delta)$*-DP algorithm, and let* $\overline{\mathcal{A}}$ *be an algorithm such that on any input* $\mathcal{D} \in \mathcal{X}^n$*, we have that the total variation distance between* $\mathcal{A}(\mathcal{D})$ *and* $\overline{\mathcal{A}}(\mathcal{D})$ *is at most* $\delta'$*. Then,* $\overline{\mathcal{A}}$ *is an* $(\epsilon, \delta + 4\delta')$*-DP algorithm.*

We next recall the following well-known result on detecting the first large element in a stream.

**Lemma 2** (Theorems 3.23, 3.24, [DR14]). $\mathsf{AboveThreshold}$ *(Algorithm 4) is* $(\epsilon, 0)$*-DP. Moreover, for* $\gamma \in (0, 1)$*, let* $\alpha = \frac{8\Delta \log(\frac{2T}{\gamma})}{\epsilon}$ *and* $\mathcal{D} \in \mathcal{X}^n$*.* $\mathsf{AboveThreshold}$ *halts at time* $k \in [T + 1]$ *such that with probability* $\geq 1 - \gamma$*:* $q_t(\mathcal{D}) \leq \tau + \alpha$ *for all* $t < k$*, and* $q_k(\mathcal{D}) \geq \tau - \alpha$ *or* $k = T + 1$.

**Geometric median.** In the rest of the paper, for $R > 0$, we fix a dataset $\mathcal{D} := \{\mathbf{x}_i\}_{i \in [n]} \subset \mathbb{B}^d(R)$, i.e., with domain $\mathcal{X} := \mathbb{B}^d(R)$. Our goal is to approximate the *geometric median* of $\mathcal{D}$, i.e.,

$$\mathbf{x}_\star(\mathcal{D}) := \arg\min_{\mathbf{x} \in \mathbb{R}^d} f_\mathcal{D}(\mathbf{x}), \text{ where } f_\mathcal{D}(\mathbf{x}) := \frac{1}{n} \sum_{i \in [n]} \|\mathbf{x} - \mathbf{x}_i\| \tag{3}$$

is the average Euclidean distance to the dataset. Following e.g., [CLM+16, HSU24], we also define the quantile radii associated with our dataset $\mathcal{D}$ centered at $\bar{\mathbf{x}} \in \mathbb{R}^d$ by

$$r^{(\tau)}(\mathcal{D}; \bar{\mathbf{x}}) := \arg\min_{r \geq 0} \left\{ \sum_{i \in [n]} \mathbb{I}_{\|\mathbf{x}_i - \bar{\mathbf{x}}\| \leq r} \geq \tau n \right\}, \text{ for all } \tau \in [0, 1]. \tag{4}$$

When $\bar{\mathbf{x}}$ is unspecified, by default $\bar{\mathbf{x}} = \mathbf{x}_\star(\mathcal{D})$. In our utility analysis we will often suppress the dependence on $\mathcal{D}$ in $\mathbf{x}_\star, r^{(\tau)}$, etc., as the dataset of interest will not change. In the privacy analysis, we specify the dependence of these functions on the dataset explicitly when comparing algorithms run on neighboring datasets. Finally, we include two helper results from prior work.

---

[4]In principle, $\epsilon$ can be larger than 1. However, in this paper, sample complexities are unaffected up to constants for any $\epsilon \geq 1$ if we simply obtain $(1, \delta)$-DP guarantees rather than $(\epsilon, \delta)$-DP guarantees, which are only stronger. Thus we state all results for $\epsilon \in [0, 1]$ for convenience, which simplifies some bounds.

**Lemma 3.** *Let $\mathcal{D} := \{\mathbf{x}_i\}_{i \in [n]} \subset \mathbb{R}^d$. Then, $f_{\mathcal{D}}(\mathbf{x}_\star) \geq (1 - \tau)r^{(\tau)}$ for all $\tau \in [0, 1]$.*

**Lemma 4** (Lemma 24, [CLM$^+$16])**.** *Let $\mathcal{D} := \{\mathbf{x}_i\}_{i \in [n]} \subset \mathbb{R}^d$ and let $S \subseteq [n]$ have $|S| < \frac{n}{2}$. Then,*

$$\|\mathbf{x}_\star - \mathbf{x}\| \leq \left(\frac{2n - 2|S|}{n - 2|S|}\right) \max_{i \notin S} \|\mathbf{x}_i - \mathbf{x}\|, \text{ for all } \mathbf{x} \in \mathbb{R}^d.$$

## 3 Constant-Factor Approximation

In this section, we give our first result: a fast algorithm for computing an $O(1)$-approximation to the geometric median. We first show how to estimate quantile radii up to constant factors in nearly-linear time using subsampled scores, improving Section 2.1 of [HSU24]. We then adapt a weighted variant of FriendlyCore to give a simple algorithm for approximate centerpoint computation, improving Section 2.2 of [HSU24] for large aspect ratios. Proofs from this section are deferred to Appendix C.

We first present our radius estimation algorithm.

---

**Algorithm 1** FastRadius$(\mathcal{D}, r, R, \epsilon, \delta)$

---

**Input:** $\mathcal{D} \in \mathbb{B}^d(R)^n$, $0 < r \leq R$, $(\epsilon, \delta) \in [0, 1]^2$

1: $(T, k, \tau) \leftarrow (\lceil \log_2(\frac{R}{r}) \rceil, 3\log(\frac{4T}{\delta}), 0.775n)$
2: $\hat{\tau} \leftarrow \tau + \nu_{\mathsf{thresh}}$ for $\nu_{\mathsf{thresh}} \sim \mathsf{Laplace}(\frac{6}{\epsilon})$
3: **for** $t \in [T]$ **do**
4:     **for** $i \in [n]$ **do**
5:         $S_t^{(i)} \leftarrow \mathsf{Unif}([n])^{\otimes k}$
6:         $N_t^{(i)} \leftarrow \frac{n}{k} \sum_{j \in S^{(i)}} \mathbb{I}_{\|\mathbf{x}_i - \mathbf{x}_j\| \leq r_t}$ for $r_t \leftarrow r \cdot 2^{t-1}$
7:     **end for**
8:     $q_t \leftarrow \frac{1}{n} \sum_{i \in [n]} N_t^{(i)}$
9:     $\nu_t \sim \mathsf{Laplace}(\frac{12}{\epsilon})$
10:     **if** $q_t + \nu_t \geq \hat{\tau}$ **then**
11:         **Return:** $r_t$
12:     **end if**
13: **end for**
14: **Return:** $R$

---

Algorithm 1 is an instance of AboveThreshold with $\Delta = 3$, where the queries are given on Line 8. However, our queries have random sensitivities depending on the subsampling in Line 5. Nonetheless, Chernoff bounds control this sensitivity with high probability, which yields privacy via Lemma 2.

**Lemma 5.** *Algorithm 1 is $(\epsilon, \delta)$-DP.*

The proof of Lemma 5 is a simple combination of Fact 4, Lemma 1, and Lemma 2. Next, we state a utility bound for Algorithm 1, whose proof is essentially identical to the proof in [HSU24], after applying Chernoff bounds to ensure our estimated counts are accurate.

**Lemma 6.** *Algorithm 1 runs in time $O(nd\log(\frac{R}{r})\log(\log(\frac{R}{r})\frac{1}{\delta}))$. Moreover, if $r \leq 4r^{(0.9)}$ and $n \geq \frac{2400}{\epsilon}\log(\frac{4T}{\delta})$, with probability $\geq 1 - \delta$, Algorithm 1 outputs $\hat{r}$ satisfying $\frac{1}{4}r^{(0.75)} \leq \hat{r} \leq 4r^{(0.9)}$.*

In summary, Lemmas 5 and 6 show that we can privately estimate $\hat{r}$ satisfying $\frac{1}{4}r^{(0.75)} \leq \hat{r} \leq 4r^{(0.9)}$ in nearly-linear time. The upper bound implies (with Lemma 3) that $\hat{r} = O(f_{\mathcal{D}}(\mathbf{x}_\star))$; on the other hand, the lower bound will be critically used in our centerpoint estimation procedure, Algorithm 2.

Algorithm 2 is a simplification of the FriendlyCore algorithm of [TCK$^+$22], that outputs a noisy weighted average of the dataset, where the weights $\{p_i\}_{i \in [n]}$ linearly interpolate estimated scores $f_i \in [0.5k, 0.75k]$ into the range $[0, 1]$, sending $f_i \geq 0.75k$ to 1, and $f_i \leq 0.5k$ to 0. We first make some basic observations about the points that contribute positively to the weighted combination $\bar{\mathbf{x}}$.

**Lemma 7.** *Assume that $\hat{r} \geq r^{(0.75)}$ in the context of Algorithm 2. With probability $\geq 1 - \frac{\delta}{18}$, every $i \in [n]$ that is assigned $p_i > 0$ in Algorithm 2 satisfies $\|\mathbf{x}_i - \mathbf{x}_\star\| \leq 3\hat{r}$, and $Z \geq 0.6n$.*

---

**Algorithm 2** FastCenter$(\mathcal{D}, \hat{r}, \epsilon, \delta)$

---

**Input:** $\mathcal{D} \in \mathbb{B}^d(R)^n$, $\hat{r} \in \mathbb{R}_{>0}$, $(\epsilon, \delta) \in [0, 1]^2$

1: $k \leftarrow 600 \log(\frac{18n}{\delta})$
2: **for** $i \in [n]$ **do**
3:      $S_i \leftarrow \mathsf{Unif}([n])^{\otimes k}$
4:      $f_i \leftarrow \sum_{j \in S^{(i)}} \mathbb{I}_{\|\mathbf{x}_i - \mathbf{x}_j\| \leq 2\hat{r}}$
5:      $p_i \leftarrow \min(\max(0, \frac{f_i - 0.5k}{0.25k}), 1)$
6: **end for**
7: $Z \leftarrow \sum_{i \in [n]} p_i$
8: $\xi \sim \mathsf{BoundedLaplace}(\frac{24}{\epsilon}, \frac{24}{\epsilon} \log(\frac{24}{\delta}))$
9: **if** $Z + \xi - \frac{24}{\epsilon} \log(\frac{24}{\delta}) \leq 0.55n$ **then**
10:      **Return:** $\mathbf{0}_d$
11: **end if**
12: $\bar{\mathbf{x}} \leftarrow \frac{1}{Z} \sum_{i \in [n]} p_i \mathbf{x}_i$
13: $\boldsymbol{\xi} \sim \mathcal{N}(\mathbf{0}_d, \sigma^2 \mathbf{I}_d)$, for $\sigma \leftarrow \frac{1600\hat{r}}{n\epsilon} \sqrt{\log(\frac{12}{\delta})}$
14: **Return:** $\bar{\mathbf{x}} + \boldsymbol{\xi}$

---

We remark that to make Lemma 7 compatible with the output guarantee of Lemma 6, it is enough to pass in $\hat{r} \leftarrow 4\hat{r}$ where $\hat{r}$ is the output of Algorithm 1. By inspection, this only changes the definition of the constant in Theorem 3, the main export of this section.

Lemma 7 constitutes most of our utility analysis, as all contributing points to $\hat{\mathbf{x}}$ are $O(\hat{r})$ away from $\mathbf{x}_\star$, our final desired error. We next note that whenever the algorithm does not return on Line 10, all surviving points lie in a ball of diameter $O(\hat{r})$, under a high-probability event over our subsampling. Importantly, this holds independently of any assumption on $\hat{r}$ (e.g., we do not require $\hat{r} \geq r^{(0.75)}$).

**Lemma 8.** *Suppose that it is the case that in the context of Algorithm 2, we have*

$$f_i^\star \leq 0.45k \implies f_i \leq 0.5k, \text{ and } f_i^\star \leq 0.55k \implies f_i \leq 0.6k, \tag{5}$$

*for all $i \in [n]$. If $Z > 0.55n$, there exists some $\mathbf{x} \in \mathbb{R}^d$ such that $\{\mathbf{x}_i \mid p_i > 0\}_{i \in [n]} \subseteq \mathbb{B}(\mathbf{x}, 4\hat{r})$. Moreover, the event (5) occurs with probability $\geq 1 - \frac{\delta}{9}$.*

We are now ready to prove a privacy bound on Algorithm 2. Our proof follows the original privacy analysis of FriendlyCore [TCK+22], but accounts for the failure of the "accurate subsampling" event (5) by coupling to an algorithm that always ensures (5), and then applying Lemma 1.

**Lemma 9.** *If $n \geq 20$, Algorithm 2 is $(\epsilon, \delta)$-DP.*

Combining our developments gives our constant-factor approximation to the geometric median.

**Theorem 3.** *Let $\mathcal{D} = \{\mathbf{x}_i\}_{i \in [n]} \subset \mathbb{B}^d(R)$ for $R > 0$, let $0 < r \leq 4r^{(0.9)}(\mathcal{D})$, and let $(\epsilon, \delta) \in [0, 1]^2$. Suppose that $n \geq C \cdot (\sqrt{d} \cdot \frac{\log(\frac{1}{\delta})}{\epsilon} + \frac{\log(\frac{\log(R/r)}{\delta})}{\epsilon})$, for a sufficiently large constant $C$. There is an $(\epsilon, \delta)$-DP algorithm that returns $(\hat{\mathbf{x}}, \hat{r})$ such that with probability $\geq 1 - \delta$, following notation (3),*

$$f_\mathcal{D}(\hat{\mathbf{x}}) \leq (40C' + 1) f_\mathcal{D}(\mathbf{x}_\star(\mathcal{D})), \quad \hat{r} \leq 4r^{(0.9)}, \tag{6}$$

*for a universal constant $C'$. Moreover, $\|\mathbf{x}_\star(\mathcal{D}) - \hat{\mathbf{x}}\| \leq C'\hat{r}$. The algorithm runs in time*

$$O\left(nd \log\left(\frac{R}{r}\right) \log\left(\frac{n \log(\frac{R}{r})}{\delta}\right)\right).$$

We remark that Theorem 3 actually comes with the slightly stronger guarantee that we obtain the optimal value for the geometric median objective $f_\mathcal{D}$, up to an additive error scaling as $O(r^{(0.9)})$. In general, while $r^{(0.9)} = O(f_\mathcal{D}(\mathbf{x}_\star(\mathcal{D})))$ always holds (Lemma 3), it is possible that $r^{(0.9)} \ll f_\mathcal{D}(\mathbf{x}_\star(\mathcal{D}))$ if a small fraction of outliers contributes significantly to $f_\mathcal{D}$. We also note that for datasets where we have more a priori information on the number of outliers we expect to see, we can adjust the quantile 0.9 in Theorem 3 to be any quantile $> 0.5$ by appropriately adjusting constants.

## 4 Boosting Approximations via Stable DP-SGD

In this section, we give a DP algorithm that efficiently minimizes the geometric median objective (3) over a domain $\mathbb{B}^d(\bar{\mathbf{x}}, \hat{r})$, given a dataset $\mathcal{D} := \{\mathbf{x}_i\}_{i \in [n]}$. In our final application to the geometric median problem, the optimization domain (i.e., the parameters $\bar{\mathbf{x}} \in \mathbb{R}^d$ and $\hat{r} \in \mathbb{R}_{\geq 0}$) will be privately estimated using Theorem 3, such that with high probability $\hat{r} = O(f_{\mathcal{D}}(\mathbf{x}_\star(\mathcal{D})))$ and $\|\mathbf{x} - \mathbf{x}_\star(\mathcal{D})\| \leq \hat{r}$. In the meantime, we treat the domain $\mathbb{B}^d(\bar{\mathbf{x}}, \hat{r})$ as a public input here.

Our strategy is to use a localization framework given by [FKT20], which gives a query-efficient reduction from private DP-SGD to stable DP-SGD executed in phases. Observe that outputting[5]

$$\frac{\mathbf{z} - \mathbf{x}_i}{\|\mathbf{z} - \mathbf{x}_i\|} = \nabla \|\cdot - \mathbf{x}_i\| (\mathbf{z}) \tag{7}$$

for $i \sim \mathsf{Unif}([n])$ is unbiased for a subgradient of $f_{\mathcal{D}}(\mathbf{z})$, leading to the following Algorithm 3.

---

**Algorithm 3** $\mathsf{StableDPSGD}(\mathcal{D}, \bar{\mathbf{x}}, \hat{r}, \rho, \delta, \eta, T)$

---

**Input:** $\mathcal{D} = \{\mathbf{x}_i\}_{i \in [n]} \subset \mathbb{R}^d$, $(\bar{\mathbf{x}}, \hat{r}) \in \mathbb{R}^d \times \mathbb{R}_{\geq 0}$, $\rho > 0$, $\delta \in (0,1)$, $\eta > 0$, $T = 2^K - 1 \geq n$ for $K \in \mathbb{N}$

1:   $m \leftarrow 3(\frac{T}{n} + \log(\frac{8}{\delta}))$
2:   **for** $k \in [K]$ **do**
3:     $(T^{(k)}, \eta^{(k)}, \sigma^{(k)}) \leftarrow (2^{-k}(T+1), 4^{-k}\eta, 3^{-k}\frac{(2m+1)\eta}{\sqrt{\rho}})$
4:     **if** $k = 1$ **then**
5:       $(\mathbf{z}_0^{(k)}, \mathcal{K}^{(k)}) \leftarrow (\bar{\mathbf{x}}, \mathbb{B}^d(\bar{\mathbf{x}}, \hat{r}))$
6:     **else**
7:       $(\mathbf{z}_0^{(k)}, \mathcal{K}^{(k)}) \leftarrow (\hat{\mathbf{x}}^{(k-1)}, \mathbb{B}^d(\hat{\mathbf{x}}^{(k-1)}, 2\sigma^{(k)}\sqrt{d\log(\frac{4K}{\delta})}))$
8:     **end if**
9:     **for** $0 \leq t < T^{(k)}$ **do**
10:      $\mathbf{g}_t^{(k)} \leftarrow \frac{\mathbf{z}_t^{(k)} - \mathbf{x}_i}{\|\mathbf{z}_t^{(k)} - \mathbf{x}_i\|}$ for $i \sim \mathsf{Unif}([n])$
11:      $\mathbf{z}_{t+1}^{(k)} \leftarrow \mathbf{\Pi}_{\mathcal{K}^{(k)}}(\mathbf{z}_t^{(k)} - \eta^{(k)}\mathbf{g}_t^{(k)})$
12:     **end for**
13:     $\bar{\mathbf{x}}^{(k)} \leftarrow \frac{1}{T^{(k)}}\sum_{0 \leq t < T^{(k)}} \mathbf{z}_t^{(k)}$
14:     $\boldsymbol{\xi}^{(k)} \sim \mathcal{N}(\mathbf{0}_d, (\sigma^{(k)})^2 \mathbf{I}_d)$
15:     $\hat{\mathbf{x}}^{(k)} \leftarrow \bar{\mathbf{x}}^{(k)} + \boldsymbol{\xi}^{(k)}$
16: **end for**
17: **Return:** $\hat{\mathbf{x}}^{(K)}$

---

**Remark 1.** *Several steps in Algorithm 3 are used only in the worst-case utility proof, and do not affect privacy. Practical optimizations can be made while preserving privacy guarantees, e.g., removing projections onto the changing domains $\mathcal{K}^{(k)}$ rather than $\mathcal{K}^{(1)}$, which is not used in the privacy proof.*

*One optimization we found useful in our experiments (described in Section 5) is replacing the random sampling on Line 10 with deterministic passes through the dataset in a fixed order. By doing so, we know the total number of accesses of any single element is $\leq m := \lceil \frac{T}{n} \rceil$ (rather than the high-probability estimate in Lemma 10 for the randomized variant in Algorithm 3). This lets us tighten the noise level $\sigma^{(k)}$ by a fairly significant constant factor, resulting in improved empirical performance.*

We provide a privacy bound on Algorithm 3 in Appendix D.1, and a utility bound in Appendix D.2. Our utility proof is fairly standard, and makes small modifications to the original localization analysis in [FKT20] to obtain a high-probability error guarantee using martingale arguments.

Our main technical novelty lies in our privacy proof, which is unconventional as it requires showing stability of DP-SGD on a non-smooth function (3). This contrasts with [FKT20], which assumed smoothness to achieve fast runtimes. Our main observation to this end is in Lemma 11 and Corollary 1, which show Line 11 implemented with a shared $\mathbf{x}_i$, but different starting $\mathbf{z}_t^{(k)}$, $(\mathbf{z}_t^{(k)})'$, is contractive,

---

[5]By default, if $\mathbf{x} = \mathbf{x}_i$, we let (7) evaluate to $\mathbf{0}_d$, which is a valid subgradient by first-order optimality.

unless $\mathbf{z}_t^{(k)}, (\mathbf{z}_t^{(k)})'$ were close. This yields a refined bound on the stability of each phase, allowing us to use the [FKT20] reduction for DP. Combining these pieces yields our following main result.

**Theorem 4.** *Let $\mathcal{D} = \{\mathbf{x}_i\}_{i\in[n]} \subset \mathbb{B}^d(R)$ for $R > 0$, let $0 < r \leq 4r^{(0.9)}(\mathcal{D})$, and let $(\alpha, \epsilon, \delta) \in [0,1]^3$. Suppose that $n \geq C \cdot (\frac{\sqrt{d}}{\alpha\epsilon} \log^{2.5}(\frac{\log(\frac{d}{\alpha\delta\epsilon})}{\delta}))$, for a sufficiently large constant $C$. There is an $(\epsilon, \delta)$-DP algorithm that returns $\hat{\mathbf{x}}$ such that with probability $\geq 1 - \delta$, following notation (3), $f_{\mathcal{D}}(\hat{\mathbf{x}}) \leq (1 + \alpha)f_{\mathcal{D}}(\mathbf{x}_\star(\mathcal{D}))$. The algorithm runs in time*

$$O\left(nd\log\left(\frac{R}{r}\right)\log\left(\frac{d\log(\frac{R}{r})}{\alpha\delta\epsilon}\right) + \frac{d}{\alpha^2}\log\left(\frac{\log(\frac{d}{\alpha\delta\epsilon})}{\delta}\right)\right).$$

## 5 Experiments

We present empirical evidence supporting the efficacy of our techniques.[6] We conduct experiments on Algorithm 1 (the radius estimation step of Section 3) and Algorithm 3, to evaluate how subsampled estimates and DP-SGD respectively improve the performance of our algorithm.

We do not present experiments on Algorithm 2, as our current analysis results in loose constants, which in our preliminary experimentation significantly impacted the performance of Algorithm 2 in practice. We leave optimizing the performance of this step as an important step for future work. Our Algorithm 2 and Section 2.2 of [HSU24] had essentially the same theoretical guarantees, so we find it in line with the conceptual contribution of this work to evaluate the other two components.

We use two types of synthetic datasets with outliers, described in Appendix E: GaussianCluster (used in [HSU24] as well), and HeavyTailed (a multivariate Student's $t$ distribution).

**Subsampling.** We performed experiments to show the benefit of subsampling in FastRadius (Algorithm 1) over RadiusFinder (Algorithm 1 from [HSU24]) for differentially private estimation of the quantile radius, which is the first step in differentially private estimation of the geometric median. Across a variety of settings of $n$, $d$, and aspect ratios in our GaussianCluster experiment and degrees of freedom in our HeavyTailed, both FastRadius and RadiusFinder consistently resulted in estimates $\approx 1.2\text{-}3\times$ the true quantile radius, with FastRadius's estimates consistently about 10-20% worse. However, the average wall-clock time for our algorithm was roughly $30\times$ faster, even at $n = 1000$, $d = 10$. We present plots and experimental results for this evaluation in Appendix E.

**Boosting.** We also evaluate the performance of our boosting algorithm in Section 4, compared to the baseline method from [HSU24]. We in fact evaluate three methods: (1) the baseline method, DPGD (vanilla DP gradient descent), as described in Algorithms 3 and 6, [HSU24], but with an optimized step size selected through ablation studies (cf. Appendix E); (2) StableDPSGD, i.e., our Algorithm 3 implemented as written, and (3) FixedOrderDPSGD, a variant of our Algorithm 3 with the last optimization described in Remark 1. We calibrated our noise level in FixedOrderDPSGD to ensure a fixed level of CDP via a group privacy argument, where we use that each dataset element is deterministically accessed at most $m = \lceil \frac{T}{n} \rceil$ times in FixedOrderDPSGD with $T$ iterations.

We present here a representative experiment, deferring a more comprehensive set of experiments to Appendix E. Here, we vary $n$ only, fixing all other parameters in a GaussianCluster dataset in $\mathbb{R}^{50}$. We report the performance of all three methods, plotting the passes over the dataset used by the excess error. Our error metric is $\frac{1}{\hat{r}} \cdot (f_{\mathcal{D}}(\hat{\mathbf{x}}) - f_{\mathcal{D}}(\hat{\mathbf{x}}))$, i.e., a multiple of the "effective radius" used. This is a more reflective performance metric than the corresponding multiple of $f_{\mathcal{D}}$, as our algorithms achieve this bound (cf. discussion after Theorem 3), and $f_{\mathcal{D}} \geq \hat{r}$ for our datasets due to outliers.

Throughout our experiments, we observed consistent trends to Figure 3, where our (optimized) algorithm FixedOrderDPSGD outperformed the baseline by an amount depending on $n$ (with better performance for larger $n$), and the unoptimized StableDPSGD achieved worse, but competitive, results. We remark that one limitation of our evaluation is that DPGD gradient computations can be parallelized, which leads to wall-clock time savings; we discuss this point further in Appendix E.

---

[6]Our subsampling experiments were performed on a single Google Colab CPU, and our boosting experiments were performed on a personal Apple M4 with 16GB RAM.

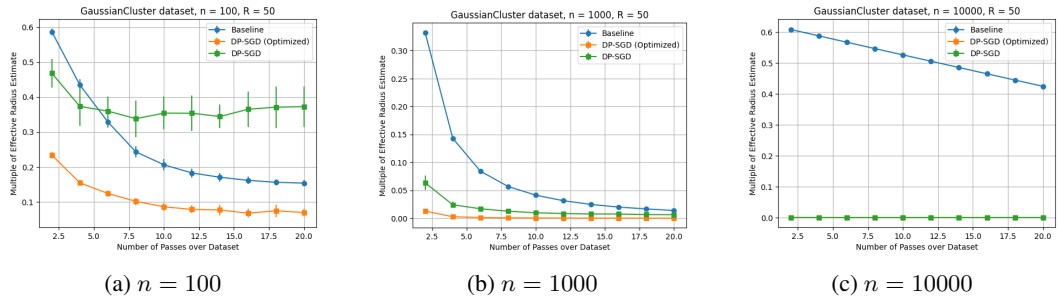

(a) $n = 100$       (b) $n = 1000$       (c) $n = 10000$

Figure 1: Comparison of DPGD, StableDPSGD, and FixedOrderDPSGD across different datasets over $\mathbb{R}^{50}$. All plots are averaged across 20 trials and standard deviations are reported as error bars.

## Acknowledgments

We gratefully acknowledge NSF grants 2217069, 2019844, and DMS 2109155. We thank the Texas Advanced Computing Center (TACC) for computing resources used in this project.

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

# A  Discussion of [HSU24] runtime

We give a brief discussion of the runtime of the [HSU24] algorithm in this section, as the claimed runtimes in the original paper do not match those described in Section 1. As the [HSU24] algorithm is split into three parts (the first two of which correspond to the warm start phase and the third of which corresponds to the boosting phase), we discuss the runtime of each part separately.

**Radius estimation.** The radius estimation component of [HSU24] corresponds to Algorithm 1 and Section 2.1 of the paper. The authors claim a runtime of $O(n^2 \log(\frac{R}{r}))$ for this step due to need to do pairwise distance comparisons on a dataset of size $n$, for $O(\log(\frac{R}{r}))$ times in total. However, we believe the runtime should be $O(n^2 d \log(\frac{R}{r}))$, accounting for the $O(d)$ cost of each comparison.

**Centerpoint estimation.** The centerpoint estimation component of [HSU24] corresponds to Algorithm 2 and Section 2.2 of the paper. We agree with the authors that this algorithm runs in time $O(nd \log(\frac{R}{r}))$, and in particular, this step does not dominate any runtime asymptotically.

**Boosting.** The boosting component of [HSU24] corresponds to Algorithms 3 and 4 and Section 3 of the paper. The authors provide two different boosting procedures (based on gradient descent and cutting-plane methods) and state their runtimes as $\widetilde{O}(n^2 d)$ and $\widetilde{O}(nd^2 + d^{2+\omega})$, where $\omega < 2.372$ is the current matrix multiplication exponent [ADV+25]. We agree with the latter runtime analysis; however, we believe there is an additive $\widetilde{O}(n^3 \epsilon^2)$ term in the runtime of gradient descent. This follows by noting that Algorithm 3 uses $\approx \frac{n^2 \epsilon^2}{d}$ iterations, each of which takes $O(nd)$ time to implement.

# B  Deferred material from Section 2

In this appendix, we provide missing material from Section 2. We begin by proving Lemmas 1 and 3.

**Lemma 1.** *For $(\epsilon, \delta) \in [0,1]^2$, $\mathcal{A} : \mathcal{X}^n \to \Omega$ be an $(\epsilon, \delta)$-DP algorithm, and let $\overline{\mathcal{A}}$ be an algorithm such that on any input $\mathcal{D} \in \mathcal{X}^n$, we have that the total variation distance between $\mathcal{A}(\mathcal{D})$ and $\overline{\mathcal{A}}(\mathcal{D})$ is at most $\delta'$. Then, $\overline{\mathcal{A}}$ is an $(\epsilon, \delta + 4\delta')$-DP algorithm.*

*Proof.* For neighboring datasets $\mathcal{D}, \mathcal{D}'$, and any event $\mathcal{E} \in \Omega$, we have that

$$
\begin{aligned}
\Pr\left[\overline{\mathcal{A}}(\mathcal{D}) \in \mathcal{E}\right] &\leq \Pr\left[\mathcal{A}(\mathcal{D}) \in \mathcal{E}\right] + \delta' \\
&\leq \exp(\epsilon) \Pr\left[\mathcal{A}(\mathcal{D}') \in \mathcal{E}\right] + \delta + \delta' \\
&\leq \exp(\epsilon) \Pr\left[\overline{\mathcal{A}}(\mathcal{D}') \in \mathcal{E}\right] + \delta + 4\delta'.
\end{aligned}
$$

The first and last lines used the assumption between $\mathcal{A}$ and $\overline{\mathcal{A}}$, and the second used that $\mathcal{A}$ is DP. □

**Lemma 3.** *Let $\mathcal{D} := \{\mathbf{x}_i\}_{i \in [n]} \subset \mathbb{R}^d$. Then, $f_{\mathcal{D}}(\mathbf{x}_\star) \geq (1 - \tau) r^{(\tau)}$ for all $\tau \in [0,1]$.*

*Proof.* This is immediate from the definition of $f_{\mathcal{D}}$ and nonnegativity of each summand $\|\cdot - \mathbf{x}_i\|$. □

Next we provide pseudocode for Algorithm 4, analyzed in Lemma 2.

---
**Algorithm 4** AboveThreshold($\mathcal{D}, \{q_t\}_{t \in [T]}, \tau, \epsilon$)

---
**Input:** $\mathcal{D} \in \mathcal{X}^n$, sensitivity-$\Delta$ queries $\{q_t : \mathcal{X}^n \to \mathbb{R}\}_{t \in [T]}, \tau \in \mathbb{R}, \epsilon > 0$

1: $\hat{\tau} \leftarrow \tau + \nu_{\mathsf{thresh}}$ for $\nu_{\mathsf{thresh}} \sim \mathsf{Laplace}(\frac{2\Delta}{\epsilon})$
2: **for** $t \in [T]$ **do**
3:     $\nu_t \sim \mathsf{Laplace}(\frac{4\Delta}{\epsilon})$
4:     **if** $q_t(\mathcal{D}) + \nu_t \geq \hat{\tau}$ **then**
5:         **Output:** $a_t \leftarrow \top$
6:         **Halt**
7:     **else**
8:         **Output:** $a_t \leftarrow \bot$
9:     **end if**
10: **end for**

---

Finally, our developments in Section 4 use the notions of Rényi DP (RDP) and central DP (CDP). We provide a self-contained summary of the definitions and properties satisfied by RDP and CDP here, but refer the reader to [BS16, Mir17] for a more detailed overview.

**Definition 2** (RDP and CDP). *Let $\alpha \geq 1$, $\rho \geq 0$. We say that a randomized algorithm $\mathcal{A}: \mathcal{X}^n \to \Omega$ satisfies $(\alpha, \rho)$-Rényi differential privacy (or, is $(\alpha, \rho)$-RDP) if for all neighboring datasets $\mathcal{D}, \mathcal{D}' \in \mathcal{X}^n$,*

$$D_\alpha(\mathcal{A}(\mathcal{D}) \| \mathcal{A}(\mathcal{D}')) \leq \alpha \rho.$$

*If this holds for all $\alpha \geq 1$, we say $\mathcal{A}$ satisfies $\rho$-central differential privacy (or, is $\rho$-CDP).*

**Fact 3** ([Mir17]). *RDP and CDP satisfy the following properties.*

1. *(Composition): If $\mathcal{A}_1: \mathcal{X}^n \to \Omega_1$ is $(\alpha, \rho_1)$-RDP and $\mathcal{A}_2: \mathcal{X}^n \times \Omega_1 \to \Omega_2$ is $(\alpha, \rho_2)$ for any fixed choice of input from $\Omega_1$, the composition of $\mathcal{A}_2$ and $\mathcal{A}_1$ is $(\alpha, \rho_1 + \rho_2)$-RDP.*

2. *(RDP to DP): If $\mathcal{A}$ is $(\alpha, \rho)$-RDP, it is also $(\alpha\rho + \frac{1}{\alpha-1}\log\frac{1}{\delta}, \delta)$-DP for all $\delta \in (0, 1)$.*

3. *(Gaussian mechanism): Let $\mathbf{v}: \mathcal{X}^n \to \mathbb{R}^k$ have sensitivity $\Delta$. Then for any $\sigma > 0$, drawing a sample from $\mathcal{N}(\mathbf{v}(\mathcal{D}), \sigma^2 \mathbf{I}_k)$ is $\frac{\Delta^2}{2\sigma^2}$-CDP.*

Finally, we require the following standard bound on binomial concentration.

**Fact 4** (Chernoff bound). *For all $i \in [n]$, let $X_i \sim \mathsf{Bern}(p_i)$ for some $p_i \in [0, 1]$, and let $\mu := \sum_{i \in [n]} p_i$ and $\hat{\mu} := \sum_{i \in [n]} X_i$. Then, $\Pr[\hat{\mu} > (1 + \epsilon)\mu] \leq \exp(-\frac{\epsilon^2 \mu}{2+\epsilon})$ for all $\epsilon \geq 0$, and $\Pr[\hat{\mu} < (1-\epsilon)\mu] \leq \exp(-\frac{\epsilon^2 \mu}{2})$ for all $\epsilon \in (0, 1)$.*

## C   Deferred material from Section 3

In this appendix, we provide missing material from Section 3.

**Lemma 5.** *Algorithm 1 is $(\epsilon, \delta)$-DP.*

*Proof.* Fix neighboring datasets $\mathcal{D}, \mathcal{D}' \in \mathbb{B}^d(R)^n$, and assume without loss that they differ in the $n^{\text{th}}$ entry. Observe that FastRadius (independently) uses randomness in two places: the random subsets in Line 5, and the Laplace noise added as in the original AboveThreshold algorithm in Lines 2 and 9.

We next claim that in any iteration $t \in [T]$, as long as the number of copies of the index $n$ occurring in $\bigcup_{i \in [n-1]} S_t^{(i)}$ is at most $2k$, then the sensitivity of the query $q_t$ is at most 3. To see this, denoting by $q_t, q_t'$ the random queries when Algorithm 1 is run on $\mathcal{D}, \mathcal{D}'$ respectively, and similarly defining $\{N_t^{(i)}, (N_t^{(i)})'\}_{i \in [n]}$, we observe that the sensitivity is controlled as follows:

$$q_t - q_t' \leq \frac{1}{n} \sum_{i \in [n-1]} \left( N_t^{(i)} - \left( N_t^{(i)} \right)' \right) + \frac{n}{n}$$

$$\leq \frac{1}{n} \cdot \frac{n}{k} \cdot \left( \text{number of copies of } n \text{ occurring in } \bigcup_{i \in [n-1]} S_t^{(i)} \right) + 1 \leq 3.$$

The first line holds because the $n^{\text{th}}$ (neighboring) point has $N_t^{(n)} \leq n$ and $(N_t^{(n)})' \geq 0$; the second is because every $\mathbb{1}_{\|\mathbf{x}_i - \mathbf{x}_j\| \leq r_t}$ used in the computation of $N_t^{(i)}$ is coupled except when $j = n$ is sampled.

Now, let $\overline{\mathcal{A}}$ denote Algorithm 1, and let $\mathcal{A}$ denote a variant that conditions on the randomly-sampled $\bigcup_{i \in [n-1]} S_t^{(i)}$ containing at most $2k$ copies of the index $n$, in all encountered iterations $t \in [T]$. By using Fact 4 (with $\mu \leftarrow k \cdot \frac{n-1}{n}$, $\epsilon \leftarrow 1$), due to our choice of $k$, $\bigcup_{i \in [n-1]} S_t^{(i)}$ contains at most $2k$ copies of $n$ except with probability $\frac{\delta}{4T}$, so by a union bound, the total variation distance between $\overline{\mathcal{A}}$ and $\mathcal{A}$ is at most $\frac{\delta}{4}$. Moreover, $\mathcal{A}$ is $(\epsilon, 0)$-DP by using Lemma 2. Thus, $\overline{\mathcal{A}}$ is $(\epsilon, \delta)$-DP using Lemma 1. $\qquad\square$

**Lemma 6.** *Algorithm 1 runs in time $O(nd\log(\frac{R}{r})\log(\log(\frac{R}{r})\frac{1}{\delta}))$. Moreover, if $r \leq 4r^{(0.9)}$ and $n \geq \frac{2400}{\epsilon}\log(\frac{4T}{\delta})$, with probability $\geq 1 - \delta$, Algorithm 1 outputs $\hat{r}$ satisfying $\frac{1}{4}r^{(0.75)} \leq \hat{r} \leq 4r^{(0.9)}$.*

*Proof.* The first claim is immediate. To see the second, for all $t \in [T]$ denote the "ideal" query by:

$$q_t^\star := \frac{1}{n} \sum_{i \in [n]} \sum_{j \in [n]} \mathbb{1}_{\|\mathbf{x}_i - \mathbf{x}_j\| \le r_t},$$

and recall $\mathbb{E} q_t = q_t^\star$. Our first claim is that with probability $\ge 1 - \frac{\delta}{2}$, the following guarantees hold for all iterations $t \in [T]$ that Algorithm 1 completes:

$$q_t^\star > 0.8n \implies q_t > 0.79n, \text{ and } q_t^\star < 0.75n \implies q_t < 0.76n. \tag{8}$$

To see the first part of (8), we can view $nq_t$ as a random sum of Bernoulli variables with mean $nq_t^\star > 0.8n^2 \ge 20000 \log(\frac{\delta}{4T})$, so Fact 4 with $\epsilon \leftarrow \frac{1}{80}$ yields the claim in iteration $t$ with probability $\ge 1 - \frac{\delta}{2T}$. Similarly, the second part of (8) follows by using Fact 4 with $\mu < 0.7n^2$ and $(1+\epsilon)\mu \leftarrow 0.71n^2$, because

$$\exp\left(-\frac{\epsilon^2 \mu}{2 + \epsilon}\right) \le \exp\left(-\frac{\epsilon^2 \mu}{151\epsilon}\right) = \exp\left(-\frac{\epsilon\mu}{151}\right) \le \exp\left(-\frac{n^2}{15100}\right) \le \frac{\delta}{2T} \tag{9}$$

for the relevant range of $n$ and $\epsilon \ge \frac{1}{75}$, $\epsilon\mu \ge \frac{n^2}{100}$. We thus obtain (8) after a union bound over all $t \in [T]$.

Now, suppose that $t \in [T]$ is the first index where $q_t + \nu_t \ge \hat{\tau}$, so that $\hat{r} = r_t$ and $\frac{1}{2}\hat{r} = r_{t-1}$, where we let $r_0 := \frac{r}{2}$. If no such query passes, then we set $t = T + 1$ by default. Then by the utility guarantees of Lemma 2, we have that with probability $\ge 1 - \frac{\delta}{2}$,

$$q_t \ge 0.76n, \ q_{t-1} \le 0.79n,$$

since $\frac{n}{100} \ge \alpha = \frac{24}{\epsilon} \log(\frac{4T}{\delta})$ . By taking the contrapositive of (8), we can conclude that with probability $\ge 1 - \delta$, we have $q_t^\star \ge 0.75n$ and $q_{t-1}^\star \le 0.8n$. Condition on this event for the rest of the proof.

Because $q_t^\star \ge 0.75n$, there is clearly some $\mathbf{x}_i \in \mathcal{D}$ such that $|\mathcal{D} \cap \mathbb{B}^d(\mathbf{x}_i, \hat{r})| \ge 0.75n$, as this is the average number of dataset elements in a radius-$\hat{r}$ ball centered at a random $\mathbf{x} \in \mathcal{D}$. Now applying Lemma 4 with $S$ set to the indices of $\mathcal{D} \setminus \mathbb{B}^d(\mathbf{x}_i, \hat{r})$, so that $|S| \le 0.25n$, gives

$$\|\mathbf{x}_\star - \mathbf{x}_i\| \le 3\hat{r} \implies \|\mathbf{x}_\star - \mathbf{x}_j\| \le 4\hat{r} \text{ for all } \mathbf{x}_j \in \mathcal{D} \cap \mathbb{B}^d(\mathbf{x}_i, \hat{r}).$$

This implies $4\hat{r} \ge r^{(0.75)}$ as claimed. Further, because $q_{t-1}^\star \le 0.8n$, we claim $\frac{\hat{r}}{2} > 2r^{(0.9)}$ cannot hold. Assume for contradiction that this happened, and let $S := \mathcal{D} \cap \mathbb{B}^d(\mathbf{x}_\star, r^{(0.9)})$. By the triangle inequality, for all of the $0.9n$ choices of $\mathbf{x}_i \in S$, we have that

$$\sum_{j \in [n]} \mathbb{1}_{\|\mathbf{x}_i - \mathbf{x}_j\| \le \frac{\hat{r}}{2}} \ge 0.9n,$$

which implies that $q_{t-1}^\star \ge 0.81n$, a contradiction. Thus, we obtain $\hat{r} \le 4r^{(0.9)}$ as well.

We remark that all of this logic handles the case where $2 \le t \le T$ is the iteration where Algorithm 1 returns. However, it is straightforward to check that the conclusion holds when $t = 1$ (i.e., $\hat{r} = r$) because we assumed $r \le 4r^{(0.9)}$, and the lower bound logic on $\hat{r}$ is the same as before. Similarly, if $\hat{r} = R$, then the upper bound logic on $\hat{r}$ is the same as before, and $2R \ge r^{(1)} \ge r^{(0.75)} \ge \frac{2}{4}r^{(0.75)}$ is clear. $\qquad\square$

**Lemma 7.** *Assume that $\hat{r} \ge r^{(0.75)}$ in the context of Algorithm 2. With probability $\ge 1 - \frac{\delta}{18}$, every $i \in [n]$ that is assigned $p_i > 0$ in Algorithm 2 satisfies $\|\mathbf{x}_i - \mathbf{x}_\star\| \le 3\hat{r}$, and $Z \ge 0.6n$.*

*Proof.* Our proof is analogous to Lemma 6, where for all $i \in [n]$ we define the "ideal score"

$$f_i^\star := \frac{k}{n} \sum_{j \in [n]} \mathbb{1}_{\|\mathbf{x}_i - \mathbf{x}_j\| \le 2\hat{r}},$$

such that $\mathbb{E} f_i = f_i^\star$. We first claim that with probability $\ge 1 - \frac{\delta}{18}$, the following hold for all $i \in [n]$:

$$f_i^\star \ge 0.75k \implies f_i \ge 0.7k, \text{ and } f_i^\star \le 0.45k \implies f_i \le 0.5k. \tag{10}$$

The first claim above is immediate from our choice of $k$ and Fact 4 (with failure probability $\leq \frac{\delta}{18n}$ for each $i \in [n]$); the second follows (with the same failure probability) similarly to (9), i.e.,

$$\exp\left(-\frac{\epsilon^2 \mu}{2+\epsilon}\right) \leq \exp\left(-\frac{\epsilon\mu}{20}\right) \leq \exp\left(-\frac{k}{400}\right) \leq \frac{\delta}{18n},$$

in our application, with $\epsilon \geq \frac{1}{9}$ and $\epsilon\mu \geq \frac{k}{20}$. Thus a union bound proves (10).

To obtain the first claim, observe that any $i \in [n]$ with $\|\mathbf{x}_i - \mathbf{x}_\star\| > 3\hat{r}$ must have that $\mathbb{B}(\mathbf{x}_i, 2\hat{r})$ does not intersect $\mathbb{B}(\mathbf{x}_\star, \hat{r})$. However, $\mathbb{B}(\mathbf{x}_\star, \hat{r})$ contains $0.75n$ points in $\mathcal{D}$ by assumption, so $f_i^\star \leq 0.25k$ and thus as long as the implication (10) holds, then $p_i = 0$ as desired. For the second claim, any $\mathbf{x}_i$ satisfying $\|\mathbf{x}_i - \mathbf{x}_\star\| \leq r^{(0.75)}$ has $|\mathbb{B}(\mathbf{x}_i, 2\hat{r}) \cap \mathcal{D}| \geq 0.75n$, so that $f_i^\star \geq 0.7k$. Thus, every such $\mathbf{x}_i$ has $p_i \geq 0.8$ as long as (10) holds, so the total contribution made by the $\geq 0.75n$ such $\mathbf{x}_i$ to $Z$ is at least $0.6n$. $\square$

**Lemma 8.** *Suppose that it is the case that in the context of Algorithm 2, we have*

$$f_i^\star \leq 0.45k \implies f_i \leq 0.5k, \text{ and } f_i^\star \leq 0.55k \implies f_i \leq 0.6k, \tag{5}$$

*for all $i \in [n]$. If $Z > 0.55n$, there exists some $\mathbf{x} \in \mathbb{R}^d$ such that $\{\mathbf{x}_i \mid p_i > 0\}_{i \in [n]} \subseteq \mathbb{B}(\mathbf{x}, 4\hat{r})$. Moreover, the event (5) occurs with probability $\geq 1 - \frac{\delta}{9}$.*

*Proof.* The first statement in (5) was proven in (10) to hold with probability $\frac{\delta}{18}$, and the second statement's proof is identical to the first half up to changing constants, so we omit it. Conditioned on this event, every $\mathbf{x}_i$ with $p_i > 0$ has $f_i^\star > 0.45k$. Moreover, because $Z > 0.55n$, there exists some $j \in [n]$ (i.e., with the maximum value of $p_j$) such that $p_j > 0.55$, which implies $f_j > 0.6k$ and thus $f_j^\star > 0.55k$.

So, we have shown that $\mathbb{B}(\mathbf{x}_j, 2\hat{r})$ contains more than $0.55n$ points in $\mathcal{D}$, and every surviving $i \in [n]$ (i.e., with positive $p_i$) contains more than $0.45n$ points in $\mathcal{D}$. Thus, $\mathbb{B}(\mathbf{x}_j, 2\hat{r})$ and $\mathbb{B}(\mathbf{x}_i, 2\hat{r})$ intersect, and in particular, $\mathbb{B}(\mathbf{x}_j, 4\hat{r})$ contains every surviving point by the triangle inequality. $\square$

**Lemma 9.** *If $n \geq 20$, Algorithm 2 is $(\epsilon, \delta)$-DP.*

*Proof.* Fix neighboring datasets $\mathcal{D}, \mathcal{D}' \in \mathbb{B}^d(R)^n$, and assume without loss that they differ in the $n^{\text{th}}$ entry $\mathbf{x}_n \neq \mathbf{x}_n'$. We will define $\mathcal{A}$, an alternate variant of Algorithm 2, which we denote $\overline{\mathcal{A}}$, where we condition on the following two events occurring. First, the index $n$ should occur at most $2k$ times in $\bigcup_{i \in [n-1]} S_i$. Second, the implications (5) must hold. It is clear that the first described event occurs with probability $\geq 1 - \frac{\delta}{18}$ by using Fact 4 with our choice of $k$, and we proved in Lemma 8 that the second described event occurs with probability $\geq 1 - \frac{\delta}{9}$. Thus, the total variation distance between $\mathcal{A}$ and $\overline{\mathcal{A}}$ is at most $\frac{\delta}{6}$. We will prove that $\mathcal{A}$ is $(\epsilon, \frac{\delta}{3})$-DP, from which Lemma 1 gives that $\overline{\mathcal{A}}$ is $(\epsilon, \delta)$-DP.

We begin by showing that according to $\mathcal{A}$, the statistic $Z + \xi - \frac{24}{\epsilon}\log\left(\frac{24}{\delta}\right)$ satisfies $(\frac{\epsilon}{2}, \frac{\delta}{6})$-DP. To do so, we will prove that $Z$ has sensitivity $\leq 12$, and then apply Fact 2. Recall that by assumption, when $\mathcal{A}$ is run the number of times $n$ appears in $\bigcup_{i \in [n]} S_i$ is at most $2k$. Thus, for coupled values of $Z, Z'$ corresponding to $\mathcal{D}, \mathcal{D}'$, where the coupling is over the random indices selected on Line 3,

$$Z - Z' \leq \frac{1}{0.25k}(k - 0) + \frac{1}{0.25k}\left(\text{number of copies of } n \text{ occurring in } \bigcup_{i \in [n-1]} S_i\right) \leq 12.$$

In fact, we note that the following stronger unsigned bound holds:

$$\sum_{i \in [n]} |p_i - p_i'| \leq 4 + \sum_{i \in [n-1]} |p_i - p_i'|$$

$$\leq 4 + \frac{1}{0.25k}\left(\text{number of copies of } n \text{ occurring in } \bigcup_{i \in [n-1]} S_i\right) \leq 12, \tag{11}$$

because the clipping to the interval $[0, 1]$ in the definitions of $p_i, p_i'$ can only improve $|p_i - p_i'|$, and the distance between the corresponding $f_i, f_i'$ is at most the number of copies of $n$ occurring in them.

Now, it remains to bound the privacy loss of the rest of $\mathcal{A}$, depending on whether Line 9 passes. If the algorithm terminates on Line 10, then there is no additional privacy loss.

Otherwise, suppose we enter the branch starting on Line 12. Our next step is to bound the sensitivity of $\bar{\mathbf{x}}$. Observe that whenever this branch is entered, we necessarily have $Z > 0.55n$ (and similarly, $Z' > 0.55n$), because $Z' + \xi - \frac{24}{\epsilon} \log(\frac{24}{\delta}) \leq Z$ deterministically. Thus, Lemma 8 guarantees that in $\mathcal{A}$, all $\mathbf{x}_i \in \mathcal{D}$ with $p_i > 0$ are contained in a ball of radius $4\hat{r}$, and similarly all surviving elements in $\mathcal{D}'$ are contained in a ball of radius $4\hat{r}$. However, there are at least $0.55n$ surviving elements of both $\mathcal{D}$ and $\mathcal{D}'$, and in particular, for the given range of $n$ there are at least two common surviving elements (one of which must be shared). A ball of radius $8\hat{r}$ around this element, which we denote $\hat{\mathbf{x}}$ in the rest of the proof, contains all surviving elements in $\mathcal{D}$ (according to $\{p_i\}_{i \in [n]}$) and in $\mathcal{D}'$ (according to $\{p_i'\}_{i \in [n]}$).

Now we wish to bound $\bar{\mathbf{x}} - \bar{\mathbf{x}}'$, where $\bar{\mathbf{x}}' := \frac{1}{Z'} \sum_{i \in [n]} p_i' \mathbf{x}_i'$. We have shown that in $\mathcal{A}$, $|Z - Z'| \leq 12$ and $\min(Z, Z') \geq 0.55n$. For convenience, define $\mathbf{y}_i := \mathbf{x}_i - \hat{\mathbf{x}}$ for all $i \in [n]$, and similarly define $\mathbf{y}_i'$. Recalling that all surviving elements of $\mathcal{D} \cup \mathcal{D}'$ are contained in $\mathbb{B}(\hat{\mathbf{x}}, 8\hat{r})$,

$$
\begin{aligned}
\|\bar{\mathbf{x}} - \bar{\mathbf{x}}'\| &= \left\| \frac{1}{Z} \sum_{i \in [n]} p_i \mathbf{y}_i - \frac{1}{Z'} \sum_{i \in [n]} p_i' \mathbf{y}_i' \right\| \\
&\leq \left| \frac{1}{Z} - \frac{1}{Z'} \right| \left\| \sum_{i \in [n-1]} p_i \mathbf{y}_i \right\| + \frac{1}{Z'} \left\| \sum_{i \in [n-1]} (p_i - p_i') \mathbf{y}_i \right\| + \frac{p_n}{Z} \|\mathbf{y}_n\| + \frac{p_n'}{Z'} \|\mathbf{y}_n'\| \\
&\leq \left| \frac{Z' - Z}{Z'} \right| \left\| \frac{1}{Z} \sum_{i \in [n-1]} p_i \mathbf{y}_i \right\| + \frac{8\hat{r}}{0.55n} \sum_{i \in [n-1]} |p_i - p_i'| + \frac{16\hat{r}}{0.55n} \\
&\leq \frac{96\hat{r}}{0.55n} + \frac{104\hat{r}}{0.55n} + \frac{16\hat{r}}{0.55n} \leq \frac{400\hat{r}}{n}.
\end{aligned}
$$

The first line shifted both $\bar{\mathbf{x}}$ and $\bar{\mathbf{x}}'$ by $\hat{\mathbf{x}}$, and the second line applied the triangle inequality. The third line applied the triangle inequality to the middle term, and bounded the contribution of $\mathbf{y}_n$ by using that $\|\mathbf{y}_n\| \leq 8\hat{r}$ if $p_n > 0$; a similar bound applies to $\mathbf{y}_n'$. In the fourth line, we used the triangle inequality on the first term, as well as that $\sum_{i \in [n-1]} |p_i - p_i'| \leq 1 + 12$ by using (11) and accounting for the $n^{\text{th}}$ point separately. Thus, $\bar{\mathbf{x}}$ has sensitivity $\frac{400\hat{r}}{n}$ in $\mathcal{A}$. Fact 1 now guarantees that Line 17 is also $(\frac{\epsilon}{2}, \frac{\delta}{6})$-DP. $\qquad \square$

**Theorem 3.** *Let $\mathcal{D} = \{\mathbf{x}_i\}_{i \in [n]} \subset \mathbb{B}^d(R)$ for $R > 0$, let $0 < r \leq 4r^{(0.9)}(\mathcal{D})$, and let $(\epsilon, \delta) \in [0,1]^2$. Suppose that $n \geq C \cdot (\sqrt{d} \cdot \frac{\log(\frac{1}{\delta})}{\epsilon} + \frac{\log(\frac{\log(R/r)}{\delta})}{\epsilon})$, for a sufficiently large constant $C$. There is an $(\epsilon, \delta)$-DP algorithm that returns $(\hat{\mathbf{x}}, \hat{r})$ such that with probability $\geq 1 - \delta$, following notation (3),*

$$
f_{\mathcal{D}}(\hat{\mathbf{x}}) \leq (40C' + 1) f_{\mathcal{D}}(\mathbf{x}_\star(\mathcal{D})), \; \hat{r} \leq 4r^{(0.9)}, \tag{6}
$$

*for a universal constant $C'$. Moreover, $\|\mathbf{x}_\star(\mathcal{D}) - \hat{\mathbf{x}}\| \leq C'\hat{r}$. The algorithm runs in time*

$$
O\left( nd \log\left(\frac{R}{r}\right) \log\left(\frac{n \log(\frac{R}{r})}{\delta}\right) \right).
$$

*Proof.* Regarding the utility bound, we will only establish that $\|\mathbf{x}_\star(\mathcal{D}) - \hat{\mathbf{x}}\| \leq C'\hat{r}$, which also gives (6) upon observing that $f_{\mathcal{D}}$ is 1-Lipschitz, and $f_{\mathcal{D}}(\mathbf{x}_\star(\mathcal{D})) \geq 0.1r^{(0.9)}(\mathcal{D})$, due to Lemma 3.

We first run Algorithm 1 with $(\epsilon, \delta) \leftarrow (\frac{\epsilon}{2}, \frac{\delta}{2})$, which gives for large enough $C$ (via Lemma 6)

$$
\frac{1}{4} r^{(0.75)} \leq \hat{r} \leq 4r^{(0.9)}, \tag{12}
$$

with probability $\geq 1 - \frac{\delta}{2}$. Next, we run Algorithm 2 with this value of $\hat{r}$, and parameters $(\epsilon, \delta) \leftarrow (\frac{\epsilon}{2}, \frac{\delta}{2})$. The privacy of composing these two algorithms now follows from Lemmas 5 and 9, and the runtime follows from Lemma 6, because Algorithm 1's runtime does not dominate upon inspection.

It remains to argue about the utility, i.e., that $\|\mathbf{x}_\star(\mathcal{D}) - \hat{\mathbf{x}}\| \le C'\hat{r}$. Conditioned on (12) holding, Lemma 7 guarantees that with probability $\ge 1 - \frac{\delta}{4}$, we have that $\|\bar{\mathbf{x}} - \mathbf{x}_\star(\mathcal{D})\| \le 3\hat{r}$, as a positively-weighted average of points in $\mathbb{B}^d(\mathbf{x}_\star(\mathcal{D}), 3\hat{r})$. Finally, for the given value of $\sigma$ in Algorithm 1, standard Gaussian concentration bounds imply that with probability $\ge 1 - \frac{\delta}{4}$,

$$\|\bar{\mathbf{x}} - \hat{\mathbf{x}}\| = \|\boldsymbol{\xi}\| \le 3\sigma\sqrt{d \log\left(\frac{4}{\delta}\right)} = O\left(\hat{r} \cdot \frac{\sqrt{d}\log(\frac{1}{\delta})}{n\epsilon}\right) = O(\hat{r}).$$

Thus, $\|\mathbf{x}_\star(\mathcal{D}) - \hat{\mathbf{x}}\| \le C'\hat{r}$ holds for an appropriate $C'$, except with probability $\delta$. $\qquad\square$

# D   Deferred material from Section 4

In this appendix, we provide a privacy and utility proof for Algorithm 3, and combine them to prove Theorem 4. To recall, Algorithm 3 proceeds in phases $K \approx \log(T)$. In each phase (loop of Lines 2 to 16) other than $k = 1$, we define a domain $\mathcal{K}^{(k)}$ centered at the output of the previous phase with geometrically shrinking radius $\propto \sigma^{(k)}$; the domain for phase $k = 1$ is simply $\mathbb{B}^d(\bar{\mathbf{x}}, \hat{r})$. After this, we take $T^{(k)}$ steps of SGD over $\mathcal{K}^{(k)}$ with step size $\eta^{(k)}$, and output a noised variant of the average iterate in Lines 13 to 15.

## D.1   Privacy of Algorithm 3

We first show that Algorithm 3 satisfies $(\epsilon, \delta)$-DP for an appropriate choice of $\rho$. When the sample functions of interest are smooth (i.e., have bounded second derivative), [FKT20] gives a proof based on the *contractivity* of iterates. This is based on the observation that gradient descent steps with respect to a smooth function are contractive for an appropriate step size (see e.g., Proposition 2.10, [FKT20]). Unfortunately, our sample functions are of the form $\|\cdot - \mathbf{x}_i\|$, which are not even differentiable, let alone smooth. Nonetheless, we inductively prove approximate contractivity of Algorithm 3's iterates by using the structure of the geometric median objective.

Throughout, we fix neighboring $\mathcal{D}, \mathcal{D}' \in (\mathbb{R}^d)^n$, and assume without loss of generality they differ in the $n^{\text{th}}$ entry. To simplify notation, we let $\mathcal{I} \in [n]^T$ denote the multiset of $T$ indices sampled in Line 10, across all phases. We prove DP of Algorithm 3 using Lemma 1, where we let $\overline{\mathcal{A}}$ denote Algorithm 3, and we let $\mathcal{A}$ denote a variant of Algorithm 3 conditioned on $\mathcal{I}$ containing at most $m := 3(\frac{T}{n} + \log(\frac{8}{\delta}))$ copies of $n$. We first bound the total variation distance between $\mathcal{A}$ and $\overline{\mathcal{A}}$ using Fact 4.

**Lemma 10.** *With probability $\ge 1 - \frac{\delta}{8}$, Algorithm 3 yields $\mathcal{I}$ containing $\le m$ copies of $n$.*

*Proof.* In expectation, we have $\frac{T}{n} \ge 1$ copies, so the result follows from Fact 4 and our choice of $m$. $\qquad\square$

We will show that $\mathcal{A}$ is $(\epsilon, \frac{\delta}{2})$-DP, upon which Lemmas 1 and 10 imply that $\overline{\mathcal{A}}$ (Algorithm 3) is $(\epsilon, \delta)$-DP. To do so, we control the sensitivity of each iterate $\mathbf{z}_t^{(k)}$, using the following two helper facts.

**Fact 5** ([Roc76]). *Let $\mathcal{K} \subset \mathbb{R}^d$ be a compact, convex set. Then for any $\mathbf{x}, \mathbf{y} \in \mathbb{R}^d$, we have*

$$\|\mathbf{\Pi}_{\mathcal{K}}(\mathbf{x}) - \mathbf{\Pi}_{\mathcal{K}}(\mathbf{y})\| \le \|\mathbf{x} - \mathbf{y}\|.$$

**Lemma 11.** *For any unit vectors $\mathbf{u}, \mathbf{v} \in \mathbb{R}^d$, and $a, b > 0$, let $\mathbf{x} = a\mathbf{u}$ and $\mathbf{y} = b\mathbf{v}$. Then, letting $\mathbf{x}' \leftarrow (a - \eta)\mathbf{u}$ and $\mathbf{y}' \leftarrow (b - \eta)\mathbf{v}$, we have $\|\mathbf{x}' - \mathbf{y}'\| \le \max(\|\mathbf{x} - \mathbf{y}\|, 3\eta)$.*

*Proof.* We claim that
$$\|\mathbf{x}' - \mathbf{y}'\| \le \|\mathbf{x} - \mathbf{y}\| \iff a + b \ge \eta, \tag{13}$$
from which the proof follows from observing that if $a + b \le \eta$, then we can trivially bound

$$\|\mathbf{x}' - \mathbf{y}'\| \le \|\mathbf{x} - \mathbf{y}\| + 2\eta \le a + b + 2\eta \le 3\eta.$$

Indeed, (13) follows from a direct expansion:

$$\|\mathbf{x} - \mathbf{y}\|^2 - \|\mathbf{x}' - \mathbf{y}'\|^2 = a^2 + b^2 - 2ab \langle \mathbf{u}, \mathbf{v} \rangle - (a - \eta)^2 - (b - \eta)^2 + 2(a - \eta)(b - \eta) \langle \mathbf{u}, \mathbf{v} \rangle$$
$$= 2\eta(a + b) - 2\eta^2 - 2\eta(a + b) \langle \mathbf{u}, \mathbf{v} \rangle + 2\eta^2 \langle \mathbf{u}, \mathbf{v} \rangle$$
$$= 2\eta(a + b - \eta)(1 - \langle \mathbf{u}, \mathbf{v} \rangle).$$

Thus, for $\eta \geq 0$ and $\langle \mathbf{u}, \mathbf{v} \rangle \geq 0$, we conclude that (13) holds. □

**Corollary 1.** *For any phase $k \in [K]$ in Algorithm 3, condition on the value of $\mathbf{z}_0^{(k)}$, and assume that $\mathcal{I}$ contains at most $m$ copies of $n$. Then for any $0 \leq t < T_k$, the sensitivity of $\mathbf{z}_t^{(k)}$ is $\leq (2m + 1)\eta^{(k)}$.*

*Proof.* Throughout this proof only, we drop the iteration $k$ from superscripts for notational simplicity, so we let $T := T^{(k)}$ and $\eta := \eta^{(k)}$, referring to the relevant iterates as $\{\mathbf{z}_t\}_{0 \leq t < T} := \{\mathbf{z}_t^{(k)}\}_{0 \leq t < T^{(k)}}$. We also refer to the index $i$ selected on Line 10 in iteration $0 \leq t < T$ by $i_t$.

Fix two copies of the $k^{\text{th}}$ phase of Algorithm 3, both initialized at $\mathbf{z}_0$, but using neighboring datasets $\mathcal{D}, \mathcal{D}'$ differing in the $n^{\text{th}}$ entry. Also, fix a realization of $\{i_t\}_{0 \leq t < T}$, such that $i_t = n$ at most $m$ choices of $t$ (note that $m$ is actually a bound on how many times $i_t = n$ across all phases, so it certainly bounds the occurrence count in a single phase). Conditioned on this realization, Algorithm 3 is now a deterministic mapping from $\mathbf{z}_0$ to the iterates $\{\mathbf{z}_t\}_{0 \leq t < T}$, depending on the dataset used.

Denote the iterates given by the dataset $\mathcal{D}$ by $\{\mathbf{z}_t\}_{0 \leq t < T}$ and the iterates given by $\mathcal{D}'$ by $\{\mathbf{z}_t'\}_{0 \leq t < T}$, so that $\mathbf{z}_0 = \mathbf{z}_0'$ by assumption. Also, let $\Phi_t := \|\mathbf{z}_t - \mathbf{z}_t'\|$ for all $0 \leq t < T$. We claim that for all $0 \leq t < T$,

$$\Phi_t \leq \max(2m_t + 1, 3)\eta, \text{ where } m_t := \sum_{0 \leq s < t} \mathbb{I}_{i_s = n}, \tag{14}$$

i.e., $m_t$ is the number of times the index $n$ was sampled in the first $t$ iterations of the phase. If we can show (14) holds, then we are done because $m_t \leq m$ by assumption.

We are left with proving (14), which we do by induction. The base case $t = 0$ is clear. Suppose (14) holds at iteration $t$. In iteration $t + 1$, if $m_{t+1} = m_t + 1$ (i.e., $i_t = n$ was sampled), then (14) holds by the triangle inequality and the induction hypothesis, because all gradient steps $\eta \mathbf{g}_t$ have $\|\eta \mathbf{g}_t\| \leq \eta$, and projection to $\mathcal{K}$ can only decrease distances (Fact 5). Otherwise, let $i_t = i \neq n$ be the sampled index, using the common point $\mathbf{x}_i \in \mathcal{D} \cap \mathcal{D}'$. Now, (14) follows from applying Lemma 11 with

$$\mathbf{x} \leftarrow \mathbf{z}_t - \mathbf{x}_i, \ \mathbf{y} \leftarrow \mathbf{z}_t' - \mathbf{x}_i, \ \mathbf{u} \leftarrow \frac{\mathbf{z}_t - \mathbf{x}_i}{\|\mathbf{z}_t - \mathbf{x}_i\|}, \ \mathbf{v} \leftarrow \frac{\mathbf{z}_t' - \mathbf{x}_i}{\|\mathbf{z}_t' - \mathbf{x}_i\|}.$$

In particular, we have that $\|\mathbf{x} - \mathbf{y}\| = \Phi_t$, and $\|\mathbf{x}' - \mathbf{y}'\| \geq \Phi_{t+1}$ (due to Fact 5), following notation from Lemma 11. We thus have $\Phi_{t+1} \leq \max(\Phi_t, 3\eta)$, which clearly also preserves (14) inductively. □

We can now conclude our privacy proof by applying composition to Corollary 1.

**Lemma 12.** *Let $\epsilon \in [0, 1]$. If $\frac{1}{\rho} \geq \frac{4 \log(\frac{2}{\delta})}{\epsilon^2} + \frac{2}{\epsilon}$, Algorithm 3 is $(\epsilon, \delta)$-DP.*

*Proof.* We claim that Algorithm 3 satisfies $\rho$-CDP, conditioned on $\mathcal{I}$ containing at most $m$ choices of $t$ (we denote this conditional variant by $\mathcal{A}$). By applying the second part of Fact 3 with $\alpha \leftarrow \frac{2 \log(\frac{2}{\delta})}{\epsilon} + 1$, this implies that $\mathcal{A}$ is $(\epsilon, \frac{\delta}{2})$-DP. Because $\mathcal{A}$ has total variation distance at most $\frac{\delta}{8}$ to Algorithm 3 due to Lemma 10, we conclude using Lemma 1 that Algorithm 3 is $(\epsilon, \delta)$-DP.

We are left to show $\mathcal{A}$ satisfies $\rho$-CDP. In fact, we will show that for all $k \in [K]$, the output of the $k^{\text{th}}$ phase of $\mathcal{A}$, i.e., $\hat{\mathbf{x}}^{(k)}$, satisfies $(\frac{16}{9})^{-k} \cdot \frac{\rho}{2}$-CDP (treating the starting iterate $\mathbf{z}_0^{(k)} = \hat{\mathbf{x}}^{(k-1)}$ as fixed). Using composition of RDP (the first part of Fact 3), this implies $\mathcal{A}$ is $\rho$-CDP as desired.

Finally, we bound the CDP of phase $k \in [K]$. Under $\mathcal{A}$, we showed in Corollary 1 that all iterates of the $k^{\text{th}}$ phase have sensitivity $\leq (2m + 1)\eta^{(k)}$. Thus the average iterate $\bar{\mathbf{x}}^{(k)}$ also has sensitivity $\leq (2m + 1)\eta^{(k)}$ by the triangle inequality. We can now bound the CDP of the $k^{\text{th}}$ phase using the third part of Fact 3:

$$\frac{((2m + 1)\eta^{(k)})^2}{2(\sigma^{(k)})^2} = \frac{((2m + 1)\eta)^2}{((2m + 1)\eta)^2} \cdot 16^{-k} \cdot 9^k \cdot \frac{\rho}{2} \leq \left(\frac{16}{9}\right)^{-k} \cdot \frac{\rho}{2}.$$

$\square$

## D.2 Utility of Algorithm 3

We now analyze the error guarantees for Algorithm 3 on optimizing the geometric median objective $f_{\mathcal{D}}$ (3). We begin by providing a high-probability bound on the utility guarantees of each single phase.

**Lemma 13.** *Following notation of Algorithm 3, we have with probability $\geq 1 - \frac{\delta}{2}$ that*

$$f_{\mathcal{D}}(\bar{\mathbf{x}}^{(1)}) - f_{\mathcal{D}}(\mathbf{x}_\star) \leq \frac{\hat{r}^2}{2\eta^{(1)}T^{(1)}} + \frac{\eta^{(1)}}{2} + 4\hat{r}\sqrt{\frac{2\log(\frac{4K}{\delta})}{T^{(1)}}},$$

*where $\mathbf{x}_\star := \operatorname{argmin}_{\mathbf{x} \in \mathbb{B}^d(\bar{\mathbf{x}}, \hat{r})} f(\mathbf{x})$, and*

$$f_{\mathcal{D}}(\bar{\mathbf{x}}^{(k)}) - f_{\mathcal{D}}(\bar{\mathbf{x}}^{(k-1)}) \leq \frac{2(\sigma^{(k)})^2 d\log(\frac{4K}{\delta})}{\eta^{(k)}T^{(k)}} + \frac{\eta^{(k)}}{2} + 8\sigma^{(k)}\log\left(\frac{4K}{\delta}\right)\sqrt{\frac{2d}{T^{(k)}}} \text{ for all } 2 \leq k \leq K.$$

*Proof.* First, with probability $\geq 1 - \frac{\delta}{4}$, we have

$$\left\|\boldsymbol{\xi}^{(k)}\right\| \leq 2\sigma^{(k)}\sqrt{d\log\left(\frac{4K}{\delta}\right)} \text{ for all } k \in [K],$$

by standard Gaussian concentration. Thus, $\bar{\mathbf{x}}^{(k-1)} \in \mathcal{K}^{(k)}$ for all $2 \leq k \leq K$, and $\mathbf{x}_\star \in \mathcal{K}^{(1)}$, except with probability $\frac{\delta}{4}$. Next, consider the $k^{\text{th}}$ phase of Algorithm 3, and for some $0 \leq t < T_k$, let us denote

$$\tilde{\mathbf{g}}_t^{(k)} := \frac{\mathbf{z}_t^{(k)} - \mathbf{x}_i}{\left\|\mathbf{z}_t^{(k)} - \mathbf{x}_i\right\|}$$

where $i \in [n]$ is the random index sampled on Line 10 in the $t^{\text{th}}$ iteration of phase $k$. We also denote

$$\mathbf{g}_t^{(k)} := \frac{1}{n}\sum_{i \in [n]} \frac{\mathbf{z}_t^{(k)} - \mathbf{x}_i}{\left\|\mathbf{z}_t^{(k)} - \mathbf{x}_i\right\|}.$$

We observe that $\mathbb{E}[\tilde{\mathbf{g}}_t^{(k)}] = \mathbf{g}_t^{(k)}$ for any realization of the randomness in all previous iterations. Now, by the standard Euclidean mirror descent analysis, see e.g., Theorem 3.2 of [Bub15], for any $\mathbf{u} \in \mathcal{K}^{(k)}$,

$$\left\langle \eta^{(k)}\tilde{\mathbf{g}}_t^{(k)}, \mathbf{z}_t^{(k)} - \mathbf{u}\right\rangle \leq \frac{\left\|\mathbf{z}_t^{(k)} - \mathbf{u}\right\|^2}{2} - \frac{\left\|\mathbf{z}_{t+1}^{(k)} - \mathbf{u}\right\|^2}{2} + \frac{(\eta^{(k)})^2}{2}. \tag{15}$$

Here we implicitly used that $\|\tilde{\mathbf{g}}_t^{(k)}\| \leq 1$ for all choices of the sampled index $i \in [n]$. Now summing (15) for all iterations $0 \leq t < T^{(k)}$, and normalizing by $\eta^{(k)}T^{(k)}$, we obtain

$$\frac{1}{T^{(k)}}\sum_{0 \leq t < T^{(k)}} \left\langle \tilde{\mathbf{g}}_t^{(k)}, \mathbf{z}_t^{(k)} - \mathbf{u}\right\rangle \leq \frac{\left\|\mathbf{z}_0^{(k)} - \mathbf{u}\right\|^2}{2\eta^{(k)}T^{(k)}} + \frac{\eta^{(k)}}{2}.$$

Next, we claim that with probability $\geq 1 - \frac{\delta}{4}$,

$$\frac{1}{T^{(1)}}\sum_{0 \leq t < T^{(1)}} \left\langle \mathbf{g}_t^{(1)} - \tilde{\mathbf{g}}_t^{(1)}, \mathbf{z}_t^{(1)} - \mathbf{u}\right\rangle \leq 4\hat{r}\sqrt{\frac{2\log\left(\frac{4K}{\delta}\right)}{T^{(1)}}},$$

$$\frac{1}{T^{(k)}}\sum_{0 \leq t < T^{(k)}} \left\langle \mathbf{g}_t^{(k)} - \tilde{\mathbf{g}}_t^{(k)}, \mathbf{z}_t^{(k)} - \mathbf{u}\right\rangle \leq 8\sigma^{(k)}\log\left(\frac{4K}{\delta}\right)\sqrt{\frac{2d}{T^{(k)}}} \text{ for all } 2 \leq k \leq K.$$

In each case, this is because $\langle \mathbf{g}_t^{(k)} - \tilde{\mathbf{g}}_t^{(k)}, \mathbf{z}_t^{(k)} - \mathbf{u}\rangle$ is a mean-zero random variable, that is bounded (with probability 1) by twice the diameter of $\mathcal{K}^{(k)}$. Thus we can bound the sub-Gaussian parameter

of their sum, and applying the Azuma-Hoeffding inequality then gives the result. Now, finally by convexity,

$$\frac{1}{T^{(1)}} \sum_{0 \leq t < T^{(1)}} \left\langle \mathbf{g}_t^{(1)}, \mathbf{z}_t^{(1)} - \mathbf{u} \right\rangle \geq \frac{1}{T^{(1)}} \sum_{0 \leq t < T^{(1)}} f_{\mathcal{D}}(\mathbf{z}_t^{(1)}) - f_{\mathcal{D}}(\mathbf{u}) \geq f_{\mathcal{D}}(\bar{\mathbf{x}}^{(1)}) - f_{\mathcal{D}}(\mathbf{u}),$$

$$\frac{1}{T^{(k)}} \sum_{0 \leq t < T^{(k)}} \left\langle \mathbf{g}_t^{(k)}, \mathbf{z}_t^{(k)} - \mathbf{u} \right\rangle \geq \frac{1}{T^{(k)}} \sum_{0 \leq t < T^{(k)}} f_{\mathcal{D}}(\mathbf{z}_t^{(k)}) - f_{\mathcal{D}}(\mathbf{u}) \geq f_{\mathcal{D}}(\bar{\mathbf{x}}^{(k)}) - f_{\mathcal{D}}(\mathbf{u}) \text{ for all } 2 \leq k \leq K.$$

Combining the above three displays, and plugging in $\mathbf{u} \leftarrow \mathbf{x}_\star$ or $\mathbf{u} \leftarrow \bar{\mathbf{x}}^{(k-1)}$, now gives the conclusion. $\qquad \square$

By summing the conclusion of Lemma 13 across all phases, we obtain an overall error bound.

**Lemma 14.** *Following notation of Algorithm 3 and Lemma 13, we have with probability $\geq 1 - \delta$ that*

$$f_{\mathcal{D}}(\hat{\mathbf{x}}^{(K)}) - f_{\mathcal{D}}(\mathbf{x}_\star) \leq \frac{\hat{r}^2}{16\eta T} + 19\eta + 8\hat{r}\sqrt{\frac{\log(\frac{4K}{\delta})}{T}} + \frac{1314 T \eta d \log^4(\frac{8K}{\delta})}{\rho n^2}.$$

*Proof.* Throughout this proof, condition on the conclusion of Lemma 13 holding, as well as

$$\left\| \boldsymbol{\xi}^{(K)} \right\| \leq 2\sigma^{(K)} \sqrt{d \log\left(\frac{2}{\delta}\right)},$$

both of which hold with probability $\geq 1 - \delta$ by a union bound. Next, by Lemma 13,

$$f_{\mathcal{D}}(\hat{\mathbf{x}}_K) - f_{\mathcal{D}}(\mathbf{x}_\star) = f_{\mathcal{D}}(\bar{\mathbf{x}}^{(1)}) - f_{\mathcal{D}}(\mathbf{x}_\star) + \sum_{k=2}^{K} f_{\mathcal{D}}(\bar{\mathbf{x}}^{(k)}) - f_{\mathcal{D}}(\bar{\mathbf{x}}^{(k-1)}) + f_{\mathcal{D}}(\hat{\mathbf{x}}_K) - f_{\mathcal{D}}(\bar{\mathbf{x}}_K)$$

$$\leq \frac{\hat{r}^2}{2\eta^{(1)} T^{(1)}} + \frac{\eta^{(1)}}{2} + 4\hat{r}\sqrt{\frac{2\log(\frac{4K}{\delta})}{T^{(1)}}}$$

$$+ \sum_{k=2}^{K} \left( \frac{2(\sigma^{(k)})^2 d \log(\frac{4K}{\delta})}{\eta^{(k)} T^{(k)}} + \frac{\eta^{(k)}}{2} + 8\sigma^{(k)} \log\left(\frac{4K}{\delta}\right) \sqrt{\frac{2d}{T^{(k)}}} \right) + \left\| \boldsymbol{\xi}^{(K)} \right\|$$

$$\leq \frac{\hat{r}^2}{16\eta T} + \frac{\eta}{2} + 8\hat{r}\sqrt{\frac{\log(\frac{4K}{\delta})}{T}} + \frac{144 m^2 \eta d \log(\frac{4K}{\delta})}{\rho T} + \frac{12\sqrt{d} m \eta \log(\frac{4K}{\delta})}{\sqrt{\rho T}}$$

$$\leq \frac{\hat{r}^2}{16\eta T} + \frac{\eta}{2} + 8\hat{r}\sqrt{\frac{\log(\frac{4K}{\delta})}{T}} + \frac{1296 T \eta d \log^3(\frac{8K}{\delta})}{\rho n^2} + \frac{36\sqrt{dT} \eta \log^2(\frac{8K}{\delta})}{\sqrt{\rho} n}$$

$$\leq \frac{\hat{r}^2}{16\eta T} + 19\eta + 8\hat{r}\sqrt{\frac{\log(\frac{4K}{\delta})}{T}} + \frac{1314 T \eta d \log^4(\frac{8K}{\delta})}{\rho n^2}.$$

The third line used Lipschitzness of $f_{\mathcal{D}}$, the fourth summed parameters using various geometric sequences, the fifth plugged in our value of $m$, and the last split the fifth term using $2ab \leq a^2 + b^2$ appropriately. $\qquad \square$

By combining Lemmas 12 and 14 with Theorem 3, we obtain our main result on privately approximating the geometric median to an arbitrary multiplicative factor $1 + \alpha$, given enough samples.

**Theorem 4.** *Let $\mathcal{D} = \{\mathbf{x}_i\}_{i \in [n]} \subset \mathbb{B}^d(R)$ for $R > 0$, let $0 < r \leq 4r^{(0.9)}(\mathcal{D})$, and let $(\alpha, \epsilon, \delta) \in [0, 1]^3$. Suppose that $n \geq C \cdot (\frac{\sqrt{d}}{\alpha\epsilon} \log^{2.5}(\frac{\log(\frac{d}{\alpha\delta\epsilon})}{\delta}))$, for a sufficiently large constant $C$. There is an $(\epsilon, \delta)$-DP algorithm that returns $\hat{\mathbf{x}}$ such that with probability $\geq 1 - \delta$, following notation (3), $f_{\mathcal{D}}(\hat{\mathbf{x}}) \leq (1 + \alpha) f_{\mathcal{D}}(\mathbf{x}_\star(\mathcal{D}))$. The algorithm runs in time*

$$O\left( nd \log\left(\frac{R}{r}\right) \log\left(\frac{d \log(\frac{R}{r})}{\alpha\delta\epsilon}\right) + \frac{d}{\alpha^2} \log\left(\frac{\log(\frac{d}{\alpha\delta\epsilon})}{\delta}\right) \right).$$

*Proof.* We first apply Theorem 3 to compute a $(\bar{\mathbf{x}}, \hat{r})$ pair satisfying

$$\hat{r} \leq C' r^{(0.9)}(\mathcal{D}), \ \|\bar{\mathbf{x}} - \mathbf{x}_\star(\mathcal{D})\| \leq \hat{r},$$

for a universal constant $C'$, subject to $(\frac{\epsilon}{2}, \frac{\delta}{2})$-DP and $\frac{\delta}{2}$ failure probability. We can verify that Theorem 3 gives these guarantees within the stated runtime, for a large enough $C$. Next, we call Algorithm 3 with $\rho \leftarrow \frac{\epsilon^2}{32 \log(\frac{4}{\delta})}$ and $\delta \leftarrow \frac{\delta}{2}$, which is $(\frac{\epsilon}{2}, \frac{\delta}{2})$-DP by Lemma 12, so this composition is $(\epsilon, \delta)$-DP.

Denoting $\hat{\mathbf{x}} := \hat{\mathbf{x}}^{(K)}$ to be the output of Algorithm 3, Lemma 14 guarantees that with probability $\geq 1 - \frac{\delta}{2}$,

$$f_\mathcal{D}(\hat{\mathbf{x}}) - f_\mathcal{D}(\mathbf{x}_\star(\mathcal{D})) \leq \frac{\hat{r}^2}{16\eta T} + 19\eta + 8\hat{r}\sqrt{\frac{\log(\frac{16K}{\delta})}{T}} + \frac{5256 T \eta d \log^5(\frac{16K}{\delta})}{\epsilon^2 n^2},$$

for some choice of $\eta, T$ and our earlier choices of privacy parameters. Optimizing in $\eta$, we have

$$f_\mathcal{D}(\hat{\mathbf{x}}) - f_\mathcal{D}(\mathbf{x}_\star(\mathcal{D})) \leq 12\hat{r}\sqrt{\frac{\log(\frac{16K}{\delta})}{T}} + \frac{37\hat{r}\sqrt{d \log^5(\frac{16K}{\delta})}}{\epsilon n}.$$

Finally, for a large enough $C$ in the definition of $n$, and $T \geq n + \frac{57600 (C')^2 \log(\frac{16K}{\delta})}{\alpha^2}$, we obtain

$$f_\mathcal{D}(\hat{\mathbf{x}}) - f_\mathcal{D}(\mathbf{x}_\star(\mathcal{D})) \leq 12\hat{r}\sqrt{\frac{\log(\frac{16K}{\delta})}{T}} + \frac{37\hat{r}\sqrt{d \log^5(\frac{16K}{\delta})}}{\epsilon n} \leq \frac{\alpha\hat{r}}{10C'} \leq \frac{\alpha r^{(0.9)}}{10} \leq \alpha f_\mathcal{D}(\mathbf{x}_\star(\mathcal{D})).$$

The last inequality used Lemma 3. Now, the runtime follows from combining Theorem 3 and the fact that every iteration of Algorithm 3 can clearly be implemented in $O(d)$ time. $\qquad\square$

# E  Deferred material from Section 5

We first describe the datasets used for our experiments in Section 5.

GaussianCluster$(R, n, d, \sigma, \text{frac}_{\text{in}})$: This dataset is described in Appendix H of [HSU24]. We draw $n_{\text{in}} = \text{frac}_{\text{in}} n$ points i.i.d. from $\mathcal{N}(\boldsymbol{\mu}, \sigma^2 \mathbf{I}_d)$ with $\boldsymbol{\mu}$ uniform on the sphere of radius $\frac{R}{2}$, and $n_{\text{out}} = n - n_{\text{in}}$ outliers uniformly from the Euclidean ball of radius $R$.

HeavyTailed$(\nu, n, d)$: This dataset samples $n$ points in $\mathbb{R}^d$ from a zero-mean multivariate Student's $t$ distribution with identity scale and degrees of freedom $\nu$.

To avoid contamination in hyperparameter selection, every experiment is performed with a fresh dataset.

**Subsampling.** We now provide more details and results on our evaluation of Algorithm 1.

In the first experiment (Figure 2a), we set $n = 1000$, $d = 10$, inlier fraction $\text{frac}_{\text{in}} = 0.9$, standard deviation $\sigma = 0.1$, and choose an upper bound $R$ from the set $\{0.5, 1, 2, 4, 8, 10\}$. The dataset is generated as GaussianCluster$(R, n, d, \sigma, \text{frac}_{\text{in}})$ dataset. We set privacy parameters $\varepsilon = 1.0$ and $\delta = 10^{-5}$, quantile fraction $\gamma = 0.75$, and for each trial we sample $r_{\min} \sim \text{Unif}([0.005, 0.02])$ to randomly initialize our search grid. Since $\gamma < \text{frac}_{\text{in}}$, we estimate the ground-truth quantile radius $r_{\text{true}} = \sigma\sqrt{d}$, run both algorithms on this dataset, measure the estimated radius $\hat{r}$ and wall-clock runtime, and report the mean and standard deviation of the estimation ratio $\hat{r}/r_{\text{true}}$ and runtime over 100 independent trials.

In the second experiment (Figure 2b), we assess the robustness of FastRadius and RadiusFinder to heavy-tailed data. We set $n = 1000$ and $d = 10$ in the HeavyTailed$(\nu, n, d)$ dataset with varying degrees of freedom $\nu \in \{2, 4, 6, 8, 10, 12, 14, 16, 18, 20\}$. For each trial, we sample $r_{\min} \sim \text{Unif}([0.005, 0.02])$, set privacy parameters $\varepsilon = 1.0$, $\delta = 10^{-5}$ and quantile fraction $\gamma = 0.75$. We estimate the theoretical quantile radius $r_{\text{true}} = \sqrt{d F_{d,\nu}(\gamma)}$ where $F_{d,\nu}$ is the CDF of an $\text{F}(d, \nu)$ distribution, which is the Fisher F-distribution with $d$ and $\nu$ degrees of freedom, execute both algorithms, record $\hat{r}$ and runtime, and summarize the mean and standard deviation of the ratio $\hat{r}/r_{\text{true}}$ and runtime across 100 repetitions.

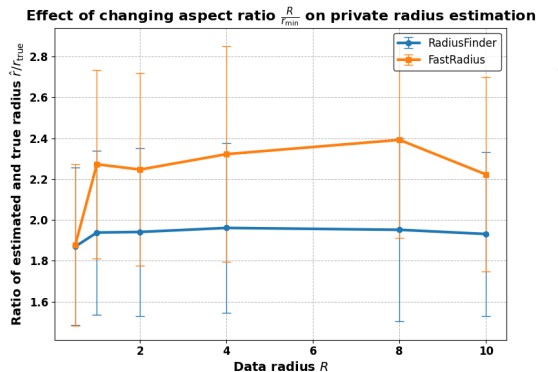
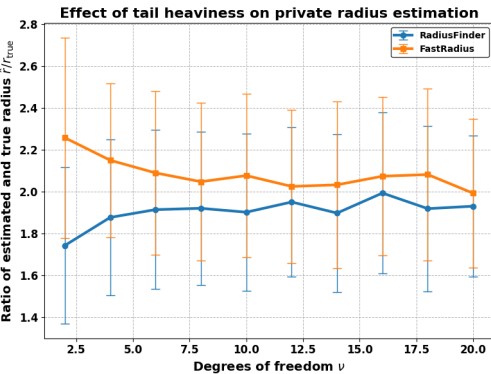

(a) Ratio of the estimated quantile radius to the true radius with varying data radius $R$.

(b) Ratio of the estimated quantile radius to the true radius with varying degrees of freedom $\nu$.

Figure 2: Comparison of RadiusFinder and FastRadius across different data distributions. All plots are averaged across 100 trials and standard deviations are reported as error bars.

We observe that in both cases, across a range of increasingly heavier tails of the distributions, both algorithms achieve reasonable approximation to the true quantile radius, always staying multiplicatively between roughly 1.2 to 3 of the true quantile radius. We further record the average wall-clock time required by both algorithms in Table 1. We observe that FastRadius is significantly faster compared to RadiusFinder, while performing competitively in terms of estimation quality.

We remark that we also experimented with varying $n \in \{500, 1000, 2000\}$ and $d \in \{5, 10, 20\}$ and observed qualitatively similar trends for the performance of both algorithms.

Table 1: Average wall-clock time in seconds over 100 trials for each algorithm in each experiment

| Experiment | RadiusFinder | FastRadius |
|---|---|---|
| Varying $R$ (Figure 2a) | $1.192 \pm 0.047$ | $0.0411 \pm 0.002$ |
| Varying $\nu$ (Figure 2b) | $1.204 \pm 0.171$ | $0.0413 \pm 0.007$ |

**Boosting.** We conclude by describing our evaluation the performance of our boosting algorithm in Section 4 based on a low-pass DP-SGD implementation, compared to the baseline method from [HSU24].

We briefly describe our hyperparameter optimization process for the baseline, DPGD. To satisfy $\rho$-CDP, Algorithm 3 in [HSU24] recommends a constant step size of $\eta_{\text{base}} = 2\hat{r}\sqrt{\frac{d}{6\rho n^2}}$, where $\hat{r}$ is the estimated radius. We examined performance using various step sizes $\eta = \eta_{\text{base}} \cdot \eta_{\text{multiplier}}$ with multipliers $\eta_{\text{multiplier}} \in \{0.5, 1, 10, 30, 50, 100\}$. Our results indicate that the $\eta_{\text{multiplier}}$ depends significantly on dataset size, in that larger datasets benefit from higher $\eta_{\text{multiplier}}$. Specifically, we observe that as $n$ increases, larger $\eta_{\text{multiplier}}$ values consistently reduce optimization error, but with diminishing returns. Based on these experiments, we select $\eta_{\text{multiplier}} = 30$ for DPGD in our experiments, as this consistently provides nearly the best performance across the range of datasets and parameter settings in our evaluation.

We now describe our setup. In all our experiments, we set $d = 50$, $\rho = 0.5$, and vary $n \in \{100, 1000, 10000\}$. For the GaussianCluster dataset, we set $\sigma = 0.1$ and vary the bounding radius $R \in \{25, 50, 100\}$. We set our estimated initial radius $\hat{r} = 20\sigma\sqrt{d}$ and initialize all algorithms at a uniformly random point on the surface of $\mathbb{B}^d(0.75\hat{r})$.[7] For the HeavyTailed dataset, we use the same values of $d$, $\rho$, and the same range of $n$. We vary $\nu \in \{2.5, 5.0, 10.0\}$, set our estimated initial radius $\hat{r} = 20\sqrt{dF_{d,\nu}(0.75)}$ to be consistent with our subsampling experiment, and again initialize randomly on the surface of $\mathbb{B}^d(0.75\hat{r})$.

---

[7]We chose a relatively pessimistic multiple of $\hat{r}$ to create a larger initial loss and account for estimation error.

In our first set of experiments, we used the middle "scale" parameter, i.e., $R = 50$ for the GaussianCluster dataset and $\nu = 5.0$ for the HeavyTailed dataset, varying $n$ only. We report the performance of the three evaluated methods, plotting the passes over the dataset used by the excess error. Our error metric is $\frac{1}{\hat{r}} \cdot (f_{\mathcal{D}}(\hat{\mathbf{x}}) - f_{\mathcal{D}}(\tilde{\mathbf{x}}))$, i.e., a multiple of the "effective radius" used in the experiment. This is a more reflective performance metric than the corresponding multiple of $f_{\mathcal{D}}$, as our algorithms achieve this bound (see discussion after Theorem 3), and $f_{\mathcal{D}} \geq \hat{r}$ for our datasets due to outliers.

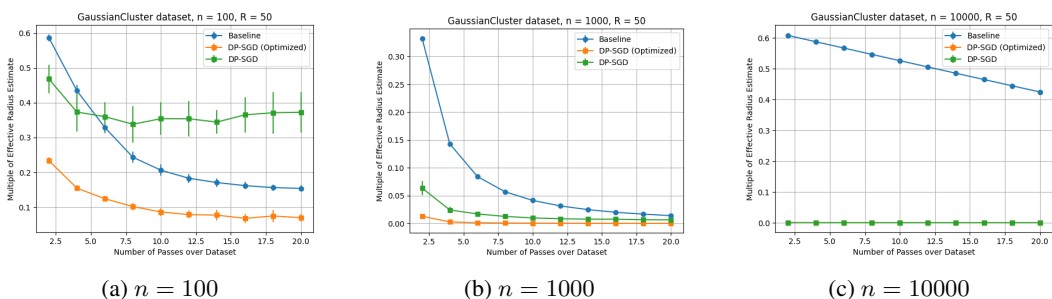

(a) $n = 100$      (b) $n = 1000$      (c) $n = 10000$

Figure 3: Comparison of DPGD, StableDPSGD, and FixedOrderDPSGD across GaussianCluster data over $\mathbb{R}^{50}$, varying $n$. Plots averaged across 20 trials and standard deviations are reported as error bars.

Across GaussianCluster datasets of size $n \in \{100, 1000, 10000\}$, we consistently saw that our optimized FixedOrderDPSGD performed the best, with comparisons between the baseline DPGD and StableDPSGD fluctuating depending on the dataset size. As the theory predicts, the gains of stochastic methods in terms of error-to-pass ratios are more stark when dataset sizes are larger, reflecting the superlinear gradient query complexity (each requiring one pass) that DPGD needs to obtain the optimal utility.

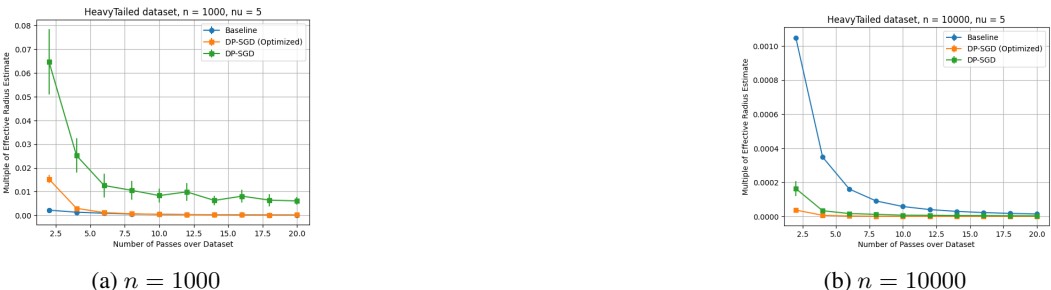

(a) $n = 1000$                 (b) $n = 10000$

Figure 4: Comparison of DPGD, StableDPSGD, and FixedOrderDPSGD across HeavyTailed data over $\mathbb{R}^{50}$, varying $n$. Plots averaged across 20 trials and standard deviations are reported as error bars.

We next present our comparisons for the HeavyTailed dataset. Here, a step size multiple of $\eta_{\text{multiplier}} = 30$ was too aggressive for the baseline DPGD to reliably converge when $n = 100$, but smaller steps impacted the performance of DPGD with larger sample sizes. On the sample sizes where DPGD successfully converged, we saw that our FixedOrderDPSGD consistently outperformed StableDPSGD, whereas the baseline DPGD slightly outperformed FixedOrderDPSGD when $n = 1000$ was moderate. The improved performance of DPGD may be related to the relative simplicity of this dataset, where all the data is distributed in a rotationally symmetric way around the "population geometric median," i.e., the mean.

In our second set of experiments, we fixed the size of the dataset at $n = 1000$, varying the scale parameter ($R$ for GaussianCluster and $\nu$ for HeavyTailed). We observed that the relative performance of our evaluated algorithms was essentially unchanged across the parameter settings we considered.

Finally, we remark that one major limitation of our evaluation is that full-batch gradient methods such as DPGD can be implemented with parallelized gradient computations, leading to wall-clock time savings. In our experiments, DPGD often performed better than FixedOrderDPSGD in terms of wall-clock time (for the same estimation error), even when it incurred significantly larger pass

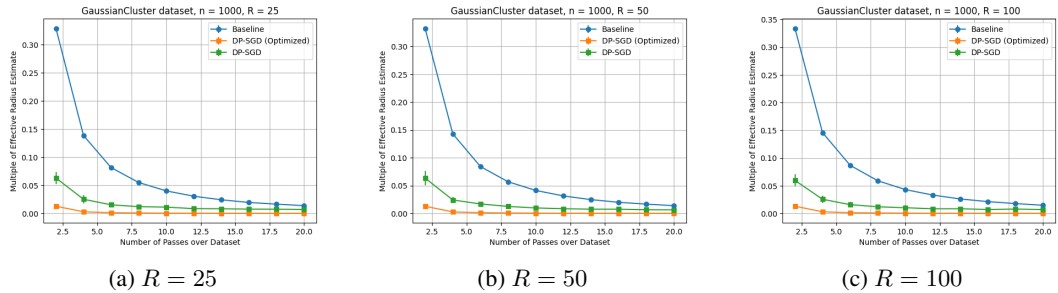

(a) $R = 25$  (b) $R = 50$  (c) $R = 100$

Figure 5: Comparison of DPGD, StableDPSGD, and FixedOrderDPSGD across GaussianCluster data over $\mathbb{R}^{50}$, varying $R$. Plots averaged across 20 trials and standard deviations are reported as error bars.

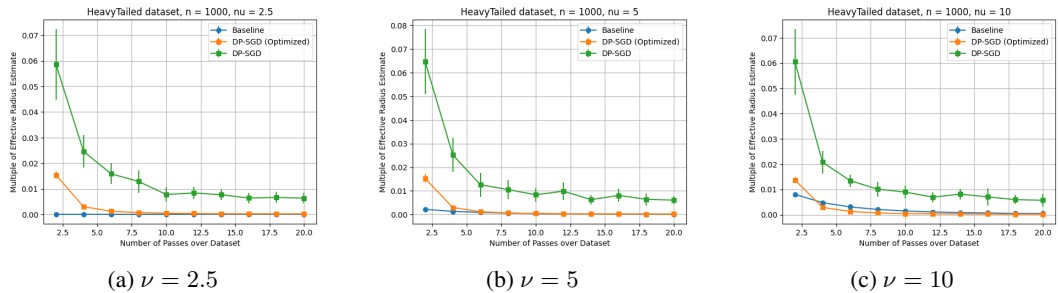

(a) $\nu = 2.5$  (b) $\nu = 5$  (c) $\nu = 10$

Figure 6: Comparison of DPGD, StableDPSGD, and FixedOrderDPSGD across HeavyTailed data over $\mathbb{R}^{50}$, varying $\nu$. Plots averaged across 20 trials and standard deviations are reported as error bars.

complexities. On the other hand, we expect the gains of methods based on DP-SGD to be larger as the dataset size and dimension $(n, d)$ grow. There are interesting natural extensions towards realizing the full potential of private optimization algorithms in practice, such as our Algorithm 3, e.g., the benefits of using adaptive step sizes or minibatches, which we believe are important and exciting future directions.

