# OpenReview forum: "Private Geometric Median in Nearly-Linear Time"
_NeurIPS.cc/2025/Conference — NeurIPS 2025 poster_

### Official Review · Reviewer_osLn · 2025-06-16

**Clarity:** 3
**Significance:** 3
**Originality:** 4
**Rating:** 5
**Confidence:** 4

**Summary:**

This paper considers the problem of privately estimating the geometric median, a topic that has been investigated previously and an information-theoretically optimal sample complexity is known. In this work, the authors propose an algorithm that achieves the same optimal sample complexity while running in a nearly linear time $\tilde{O}(nd + d/\alpha^2)$ for $n$ input data points of dimension $d$, significantly improving the previous result of [HSU24]. The workflow closely follows the framework established by [HSU24], incorporating several novel techniques that enhance runtime efficiency.

[HSU24] Mahdi Haghifam, Thomas Steinke, and Jonathan Ullman. Private geometric median. Advances in Neural Information Processing Systems, 37:46254–46293, 2024.

**Questions:**

1. Is it possible to remove the requirement of knowledge $r$ and $R$ ? As mentioned in the paper, the resulting sample complexity is independent of these parameters. I wonder if this dependence in the time complexity bound can be removed as well.
2. This work focuses on approximate DP, where a small probability $\delta$ of privacy violation is allowed. Is it able to achieve the more stringent notion of pure DP?
3. The runtime of the private algorithm is $\tilde{O}(nd + d/\alpha^2)$ while one of the non-private algorithms in [CLM+16] runs in $O(d/\alpha^2)$. Is it possible to develop a private algorithm with this time complexity?

Some (possible) typos:

1. [Line 110] "the FriendlyCore algorithm of [HSU24]": This seems to be a wrong citation.
2. In the proof of Lemma 5 [Line 520], the setting of $\mu$ and $\epsilon$ may only lead to a probability bound of $\exp(-k\cdot \frac{n-1}{n} /3)$, which is slightly looser than the claimed $\delta / 4T$.
3. [Line 537] Lemma 6 should be referenced as Lemma 2.
4. [Line 563] $f_i^\star\ge 0.7 k$ should be corrected to $f_i^\star \ge 0.75 k$
5. [Line 618] what is meant by "large enough $C$"?
6. [Line 619] "we run Algorithm 2 with this value of $\hat{r}$." However, Lemma 7 requires $\hat{r} \ge r^{(0.75)}$ while Lemma 6 only guarantees $\hat{r}\ge \frac{1}{4}r^{(0.75)}$. I think this should be corrected by passing $4\hat{r}$ to Algorithm $2$, and some related quantities need to be modified accordingly.
7. [Line 623] Lemma 6 should be referenced as Lemma 7.

[CLM+16] Michael B Cohen, Yin Tat Lee, Gary Miller, Jakub Pachocki, and Aaron Sidford. Geometric median in nearly linear time. In Proceedings of the forty-eighth annual ACM symposium on Theory of Computing, pages 9–21, 2016.

**Ethical Concerns:**

["NO or VERY MINOR ethics concerns only"]

**Final Justification:**

The rebuttal has addressed my concerns. I would like to keep my positive score.

**Limitations:**

yes

**Quality:**

4

**Strengths And Weaknesses:**

Strengths:

1. The estimation of the geometric median is a fundamental problem in computational geometry. This work presents a private algorithm that operates in nearly linear time, marking a substantial advancement over previous results.
2. The desired runtime is achieved by making several improvements to different parts of [HSU24]'s framework, which are novel and technically solid.
3. Comprehensive experiments are conducted to show the effectiveness of the proposed techniques.

Weakness:

1. The algorithm requires prior knowledge of parameters $r$ and $R$, which provides lower and upper bounds on the effective radius $r^{(0.9)}$.
2. In Section 5 the authors evaluate only Algorithm 1 and 3. The presented analysis of Algorithm 2 incurs large constants that largely affect its performance in practice. Therefore, the entire algorithm is not ready for real-world deployment.

---

> ### Author Rebuttal · Authors · 2025-07-30
>
> Thank for your encouraging words about our contribution and runtime efficiency. We will fix all typographical errors pointed out by the reviewer in the revised manuscript and address key questions below:
>
> **Knowledge of $r$ and $R$**: Thank you for the interesting question. The reason for having these constants in the algorithm are due to application of the AboveThreshold technique, a standard method in DP, in Algorithm 1 which requires a finite set of queries and we are currently not aware of extensions to this method which can handle potentially unbounded range of queries. Moreover, [BNSV15] shows that some knowledge of the parameter $r$ is necessary (as observed by prior work [HSU24]), although it is an interesting open question to improve our dependence.
>
> **On Pure DP**: Currently, the subsampling based approach used in Algorithm 1 and 2 requires a failure probability, which is critical for the computational speed-up. Furthermore, the tools used for private stochastic convex optimization in [FKT20], which inspire the analysis of Algorithm 3 in our work also currently provide an approximate DP guarantee. Therefore, generalizing both of these techniques to handle pure DP remains an interesting direction for future work. We would like to point out that the dependence of both the runtime and the sample complexity on $1/\delta$ is polylogarithmic, and therefore allows for $\delta$ being polynomially small in $1/n$.
>
> **On runtime comparison with [CLM+16]**: A private algorithm cannot run in time $\widetilde{O}(d/\alpha^2)$, given that this runtime is independent of the privacy parameter eps, which shows up polynomially in the sample complexity. We do think it is interesting to remove this additive runtime term (and obtain an $\widetilde{O}(nd)$ runtime, the other runtime in [CLM+16]), although this too seems quite challenging, as existing nearly-linear time methods for DP-ERM treat the objective as a stochastic optimization problem, and thus incur the runtime overhead of (non-private) SGD.
>
> **On constants in Algorithm 2**: We acknowledge in lines 289-293 that the constants in Algorithm 2 can potentially be further optimized as part of future work.
>
> [BNSV15] M. Bun, K. Nissim, U. Stemmer, and S. Vadhan. “Differentially private release and learning of threshold functions”. In: 2015 IEEE 56th Annual Symposium on Foundations of Computer Science. IEEE. 2015, pp. 634–649.

---

> > ### Comment · Reviewer_osLn · 2025-08-02
> >
> > Thank you for your response.

---

### Official Review · Reviewer_Nqyz · 2025-07-03

**Clarity:** 4
**Significance:** 3
**Originality:** 3
**Rating:** 5
**Confidence:** 2

**Summary:**

The task is to compute geometric median of a point set in a differentially private way. The geometric median is the minimizer of $\sum_i ||\cdot - x_i ||$ for Euclidean norm, and this paper is happy with approximate minimizer achieving $1+\alpha$  multiple of the minimum.

A recent work (HSU24) has provided an algorithm solving the task with a certain sample complexity and proved that it is the optimal complexity. However, there was a significant gap in the computational efficiency between private and non-private geometric median computation in the prior work.

This work provides an algorithm similar in spirit to the one of (HSU24) that runs in a very similar time to the non-private algorithms. Per my understanding, the differences in the algorithms are subtle (but necessary for the improved runtime), but the big picture stays the same.

The algorithm runs in three phases, first, an effective radius r (i.e., the majority of points are contained in a ball of this radius) of the input is computed, then, a crude estimate of the geometric median (in a distance at most r) is computed. Finally, a custom DP-SGD analysis is performed leveraging the structure of geometric median problem.

**Questions:**

-

**Ethical Concerns:**

["NO or VERY MINOR ethics concerns only"]

**Final Justification:**

I still think the paper is good and should be accepted.

**Limitations:**

-

**Paper Formatting Concerns:**

-

**Quality:**

4

**Strengths And Weaknesses:**

The paper is easy to read and I like the way how is it written, that it provides the intuition and sketches results before diving into the technicalities. I would only suggest to introduce the $r^{(0.9)}$ notation more explicitly.

I find this to be a valuable contribution since (private) geometric median problem is quite important and the improvements in runtime are significant. The DP-SGD analysis technique used in the paper taking advantage of the fact that the gradients are unit vectors can potentially also be used elsewher.

On the other hand, the scope of the paper is very narrow and the improvements are only in the run time and that is why I do not lean to the highest score.

---

> ### Author Rebuttal · Authors · 2025-07-30
>
> Thank you for the positive assessment. We will insert an explicit definition of the “$r^{(0.9)}$” notation where it first appears and polish notation throughout.

---

> > ### Comment · Reviewer_Nqyz · 2025-08-01
> >
> > Thank you for your response.

---

### Official Review · Reviewer_rNu7 · 2025-07-05

**Clarity:** 4
**Significance:** 3
**Originality:** 3
**Rating:** 5
**Confidence:** 4

**Summary:**

This paper proposes a DP optimization algorithm to solve the following task: given (x1,...,xn) in (R^d)^n, such that the data points have a norm of R, we aim to solve x* = argmin_x sum_i ||x - xi||. This problem is known as private geometric median. The previous work [Haghifam-Steinke-Ullman'24] proposes an algorithm to provide (1+alpha) multiplicative error with the sample complexity of sqrt(d)/(alpha*epsilon). However, the runtime of their proposed algorithms is either quadratic in n or quadratic in d. The main result of this paper is to push the time complexity to nd + d/alpha^2. Notice that d/alpha^2 is the optimal time complexity of the first-order method even non-privately.
The idea behind the proposed algorithm is as follows: it has three stages.

1- In the first stage, an algorithm similar to the algorithm proposed in [Haghifam-Steinke-Ullman'24] (these algorithms are inspired by GoodRadius of [Stemmer-Nissim-Vadhan'16]). The goal is to find a radius privately around the geometric median that contains 70% of the data points. The main idea is instead of computing all the n^2 pairwise distances is sub-sampling that makes the algorithm run in linear time.

2- The second stage is inspired by the FriendlyCore of Tsfadia et al. and it consists of finding a center with the promise that there exists a radius that contains 70% of the data points. The new idea is that FriendlyCore can be sped up to run in O~(nd) time (independently of R/r) using subsampled weights.

3- Finally, the third stage, which is very interesting, is a bespoke analysis of the DP-SGD for the geometric median loss function. The idea here is to "warm-start" DP-SGD from the centerpoint given by the output of Stage 2 and the radius given by Stage 1. The main idea is to show that when gradient descent steps are contractive for an appropriate step size, we can use the reduction in [Koren-Feldman-Talwar'20] to convert the contractivity to a private algorithm.

**Questions:**

please see above the weakness part

**Ethical Concerns:**

["NO or VERY MINOR ethics concerns only"]

**Final Justification:**

I think this paper merits acceptance.

**Limitations:**

yes

**Quality:**

3

**Strengths And Weaknesses:**

The result of this paper is interesting: it combines lots of interesting techniques to provide a very intuitive algorithm that improves the prior work's runtime.

There are a few weaknesses:

- If we view this problem from the standpoint of DP optimization, it would be interesting to find a broader class of loss functions for which we can obtain an algorithm with multiplicative guarantee.

- The optimal dependence of the sample complexity on R is unclear. Note that in one dimension there are algorithms that require log*(R) samples.

- The problem of geometric median is very related to the problem of privately finding a point in the convex hull. A helpful comparison for the community is to compare the problem of private geometric median with the problem of privately finding a point in the convex hull, for instance, see the following paper:
Kaplan, H., Mansour, Y., Moran, S., Stemmer, U., & Tur, N. (2025, June). On differentially private linear algebra. In Proceedings of the 57th Annual ACM Symposium on Theory of Computing (pp. 2362-2373).

Line 110 - the citation is wrong for the FriendlyCore algorithm.

---

> ### Author Rebuttal · Authors · 2025-07-30
>
> We thank the reviewer for their positive words about our techniques and improvement in runtime. We will fix the citation in Line 110 to replace it with [TCK+21]; great catch.
>
> **Re: broader class with multiplicative guarantees**: We agree that this is a natural and important follow-up direction, and will mention this as an open problem in the revision. Some natural candidates are problems for which there exist non-private first-order methods with multiplicative guarantees, such as other GLMs (e.g., based on Huber loss) and approximate positive LPs.
>
> **Re: dependence on $R$**: We agree that it is unclear whether our dependence on $R$ is optimal, and that this is an interesting open question. We do note that many of the techniques used in prior work appear specific to the 1-dimensional setting, and that extending these tools to arbitrary dimension would likely come at a significant cost to dependences on other parameters.
>
> **Re: privately finding point in convex hull**: This is an interesting point of comparison. Exactly computing the geometric median privately is certainly a more challenging problem than the private convex hull problem; on the other hand, our paper only aims to solve an approximate version. Current techniques for private convex hull appear substantially more complex than for approximate private geometric median; as noted by [KMM+25], private convex hull can be very unstable as a function of the dataset. On the other hand, because our goal is only to approximate the optimum multiplicatively (which scales at least linearly with $r^{(\alpha)}$ for any constant alpha > 0), it cannot be affected substantially by moving a single point, so our problem is potentially much simpler. We will mention this connection in a revision, and find it interesting to further explore the relationship between these two problems.

---

### Official Review · Reviewer_8pT7 · 2025-07-08

**Clarity:** 4
**Significance:** 4
**Originality:** 4
**Rating:** 5
**Confidence:** 4

**Summary:**

This paper presents a novel and practically relevant algorithm for computing a differentially private geometric median in high dimensions, achieving nearly-linear runtime. The authors build upon the FriendlyCore warm-start technique and integrate a refined DP-SGD procedure with a smoothed subgradient to overcome the non-smoothness of the geometric median objective.
Their proposed algorithm achieves:
- Near-optimal sample complexity
- Strong error guarantees comparable to prior work
- A significant improvement in runtime from prior quadratic-time algorithms to near-linear.
Empirical evaluation demonstrates a 30 times speedup on synthetic and real data benchmarks, without significant loss of accuracy, and with robust error performance across multiple trials.

**Questions:**

The algorithm depends on the parameters r, R? The upper bound is more justified as there may be a natural bound on the points or one could normalize them. However, r is less justified. How can one compute it and what happens if it is not correct? Could you obtain an error with an additive term relative to r?

- Lemma 9 seems strange. How does n >= 20 relate to (eps,delta)-dp? Seems like the condition is artificial and It would be nice to remove it.

- I am never a fan of misusing the O tilde notation. Theorem 1 writes O tilde to hide all polylogarithmic factors (even for unrelated variables) but it should only be used to mean hiding polylogarithmic factors of the quantity inside the O tilde. It would be nice to fix the theorems appropriately rather than adding a footnote.

**Ethical Concerns:**

["NO or VERY MINOR ethics concerns only"]

**Limitations:**

Yes, the authors describe the potential suboptimality of the runtime as a limitation tied to using first order methods for optimization. Alternative methods could potentially have better dependency.

**Paper Formatting Concerns:**

No concerns

**Quality:**

4

**Strengths And Weaknesses:**

The geometric median problem is a fundamental problem in many domains including clustering, robust statistics and social choice. This paper makes a technically sound and valuable contribution offering a novel and efficient . The combination of a warm-start approximation with a customly analyzed DP-SGD routine is elegant and well-justified, addressing the challenge of non-smooth optimization under differential privacy.

The authors do a rigorous analysis using mathematical proofs but also give an efficient implementation and empirical evaluation of their method. The writing is clear and well-organized, with helpful figures (notably Figures 1 and 2) to illustrate empirical gains.

---

> ### Author Rebuttal · Authors · 2025-07-30
>
> We thank the reviewer for their encouraging words regarding our contribution and rigorous analysis.
>
> **Need for parameters $r, R$** : As the reviewer mentions, $R$ can be estimated by a norm bound on the input datapoints. Regarding the need for estimating r, as noted in the prior work [HSU24], knowledge of such an a priori lower bound is necessary by the impossibility results in [BNSV15]. We note that our dependence on $r$ is relatively mild (i.e., essentially a single logarithmic factor in the runtime and a single loglog factor in the sample complexity). It is an interesting open question to improve the dependence on $R/r$, although we find it potentially challenging to do so while retaining the nearly-linear runtime.
>
> **Lemma 9 and $n \geq 20$**: The $n \geq 20$ is due to the particular constants used in our application of Chernoff bounds in Lemma 8 and we will optimize it in the revised manuscript.
>
> **$\widetilde{O}$ notation**: We will modify $\widetilde{O}$ to only include polylogarithmic factors of the quantity inside it, as our use may be confusing re: the dependence on $R/r$. We do note that we are not aware of a uniform convention in the usage of $\widetilde{O}$, with many related works using a similar definition to our paper. Also, we remark this notation is only used in Section 1; statements in all other sections fully quantify dependences on all parameters.
>
> [BNSV15] M. Bun, K. Nissim, U. Stemmer, and S. Vadhan. “Differentially private release and learning of threshold functions”. In: 2015 IEEE 56th Annual Symposium on Foundations of Computer Science. IEEE. 2015, pp. 634–649.

---

> > ### Comment · Reviewer_8pT7 · 2025-08-05
> >
> > Thank you for your response. Indeed, making the dependencies more explicit would be nice and make it easier for follow-up work to use and extend the resutls.

---

### Official Review · Reviewer_bdcd · 2025-07-19

**Clarity:** 3
**Significance:** 3
**Originality:** 4
**Rating:** 5
**Confidence:** 3

**Summary:**

This paper presents a new efficient algorithm for computing in a
differentially private way the geometric median of a set of
data. Recent work has proposed two-phases (warm-up and boosting)
algorithms for a multiplicative approximation of the geometric median
with optimal sample complexity but running in polynomial time. The
algorithm proposed in this paper achieve similar accuracy and sample
complexity but runs in almost linear time (for some range of the
parameters).

**Questions:**

Can you clarify why you use the FriendlyCore procedure for estimating the centerpoints? What do you gain?

**Ethical Concerns:**

["NO or VERY MINOR ethics concerns only"]

**Limitations:**

yes

**Quality:**

4

**Strengths And Weaknesses:**

Strengths

- The paper address an important problem: improving the runtime of
 algorithms for differentially private geometric median with good
 average error rates.

- The paper improves the runtime of the warm up phase by using
  subsampling in a

- In the boosting phase the paper gives an analysis of DPSGD
  leveraging the properties of the geometric median to obtain an
  improved runtime.


Weaknesses

- The proposed algorithm uses as a subroutine an adaptation of the
  FriendlyCore procedure from [TCK+22] but it is unclear why this
  departure from the original algorithm is needed.

Recent work proposed differentially private algorithms for the
geometric median with average error guarantees but those algorithm
have a polynomial runtime, which can be problematic in large data
settings. It is thus important to have more efficient algorithms and
this paper does exactly that.The proposed algorithm uses a two-phases
approach, based on warm-up and boosting, as the ones in previous work
but it optimizes the running time of each step.

To optimize the runtime of the warm-up, which has itself two phases,
one for estimating the radius and one for estimating the centerpoints,
the paper uses subsampling. In the first phase the algorithm subsample
points to estimate the radius. This change is relatively
straightforward.  In the second phase, the proposed algorithm departs
from previous work and uses as a subcomponent the FriendlyCore
procedure from [TCK+22].  In orde to improve the efficiency of this
procedure the algorithm uses subsample in the computation of the
weights associated with each data point. This change is non-trivial
and needs an interesting change in the privacy proof. However, it is
unclear why this departure from the original algorithm is needed since
the obtained improvement is quite limited.
To optimize the runtime of the boosting phase the paper proposes an
interesting ad-hoc analysis of DP-SGD specific to the geometric
median.  Specifically, the paper uses the properties of the geometric
median to control the sensitivity of DP-SGD and obtain an optimization
in nearly-linear time.
Overall, the proposed algorithm shows considerable improvement thanks
to non-trivial changes to the algorithms from previous work.

A couple of minor points:
- in equation (2) the notation for the indicator function is not yet
  defined
- line 110 has a wrong citation

---

> ### Author Rebuttal · Authors · 2025-07-30
>
> We thank the reviewer for their kind words regarding the importance of the problem considered and the improved runtime. We will add the definition of the indicator in Eq. (2) and fix the citation in Line 110 to replace it with [TCK+21] – thank you for bringing up these comments.
>
> We agree that there is no significant qualitative change in our runtime vs. that of [HSU24] in the centerpoint estimation phase. We stated as such in Section 1.2, but can further clarify that our motivation for including our alternative approach was to highlight the simplicity of directly adapting FriendlyCore. Indeed, our analysis of Algorithm 2 essentially directly extends [TCK+22] to tolerate inaccuracies due to subsampling. Our approach thus
>
> (1) gives an arguably much simpler approach to centerpoint estimation (prior work [HSU24] required a custom analysis of DP-GD with geometrically decaying step sizes), and
>
> (2) makes the observation that FriendlyCore can be sped up via subsampling, which to our knowledge has not been explicitly noted before. We do not view this section as a major technical contribution of our paper.

---

> > ### Comment · Reviewer_bdcd · 2025-08-01
> >
> > Thank you for your response. integrating part of this response in the paper could help clarity.

---

### Decision · Program_Chairs · 2025-09-17

**Decision:**

Accept (poster)

**Comment:**

This paper presents a new, efficient algorithm for computing the geometric median of a dataset in a differentially private manner. The proposed algorithm achieves a near-linear runtime, a significant improvement over prior polynomial-time works, while maintaining similar accuracy and sample complexity. The reviewers agree that the paper addresses an important problem and that the results are significant. The improved time complexity is a strong and clear technical contribution. Given the unanimous support and the paper's strengths, my recommendation is to ACCEPT.